# Learning Diverse Policies in MOBA Games via Macro-Goals

**Yiming Gao**[1]   **Bei Shi**[1]   **Xueying Du**[1]   **Liang Wang**[1]   **Guangwei Chen**[1]
**Zhenjie Lian**[1]   **Fuhao Qiu**[1]   **Guoan Han**[1]   **Weixuan Wang**[1]
**Deheng Ye**[1]   **Qiang Fu**[1]   **Wei Yang**[1]   **Lanxiao Huang**[2]
[1]Tencent AI Lab, Shenzhen, China
[2]Tencent TiMi L1 Studio, Chengdu, China
{yatminggao,beishi,sherinedu,enginewang,gorvinchen,
leolian,frankfhqiu,guoanhan,waihinwang,dericye,
leonfu,willyang,jackiehuang}@tencent.com

## Abstract

Recently, many researchers have made successful progress in building the AI systems for MOBA-game-playing with deep reinforcement learning, such as on Dota 2 and *Honor of Kings*. Even though these AI systems have achieved or even exceeded human-level performance, they still suffer from the lack of policy diversity. In this paper, we propose a novel **M**acro-**G**oals **G**uided framework, called MGG, to learn diverse policies in MOBA games. MGG abstracts strategies as macro-goals from human demonstrations and trains a Meta-Controller to predict these macro-goals. To enhance policy diversity, MGG samples macro-goals from the Meta-Controller prediction and guides the training process towards these goals. Experimental results on the typical MOBA game *Honor of Kings* demonstrate that MGG can execute diverse policies in different matches and lineups, and also outperform the state-of-the-art methods over 102 heroes.

## 1 Introduction

As a stepping stone of Artificial General Intelligence (AGI), Game AI has achieved significant progress in the past decade, including board games Silver *et al.* [2016], poker games Brown *et al.* [2019]; Moravčík *et al.* [2017], Real-time Strategy (RTS) games Vinyals *et al.* [2019], etc. Especially, Multi-player Online Battle Arena (MOBA) games have been regarded as a challenging problem in Game AI and attracted massive attention from research communities because of the complicated mechanisms such as imperfect information, large state-action space, and long-term sequential decision making. Recently, OpenAI *et al.* [2019] develop the AI system on Dota 2, named OpenAI Five, which defeats the Dota 2 world champion, i.e., Team OG, within a limited size of hero pool. It trains with deep reinforcement learning and self-play. Ye *et al.* [2020a] build the AI system on another popular MOBA game, *Honor of Kings*, which also defeats top e-sports players. Besides, compared with OpenAI Five, their AI system enables playing full MOBA games, which extends the hero pool size from 17 to 40 heroes, adds ban/pick capabilities, and addresses the scalability issue.

Even though existing AI systems have achieved or even exceeded human-level performance in MOBA games, they still suffer from the lack of policy diversity. Agents during training are controlled by the same hand-crafted reward signal and then present similar strategies even in different matches, while strategies played by humans usually vary according to situations in the game. One of the disadvantages is that we can counter the single learned policy after several attempts effortlessly. For example, one team defeats OpenAI Five nine times with the same strategy in public tests[1]. The

---

[1]https://openai.com/projects/five/

primary counter-strategy is to avoid fights as much as possible and instead push outside lanes to pressure the agents into defending[2].

Another disadvantage is that agents with a single policy cannot exploit the strength of heroes and lineups. To interest players, MOBA game designers usually make different skill mechanisms for different heroes, which result in heroes play different roles in the game. Therefore, different roles in lineups have different strategies to maximize a team's utility. For example, the supporter should keep their allies alive and give them opportunities to earn more gold and experience. The assassin can efficiently jungle neutrals rather than farm lanes. Moreover, different combinations of heroes construct lineups with different characteristics. Some lineups have advantages in pushing with a quick pace, while some lineups are good at the development of carry heroes. The agent should find a way to maximize the picked lineup's strength instead of playing with a single policy in the game.

Although existing methods can learn diverse policies in other games, it remains to be a grand challenge in MOBA games. AlphaStar Vinyals *et al.* [2019] uses a framework based on the league to improve the diversity of policies in StarCraft II. The historical models are used to construct the league and train exploiters to highlight flaws in the league and main agents. The main agents are forced to discover new strategies by training against the exploiters. They also define a latent variable according to building orders in human data as constraints of self-play. However, the league-based methods cannot be applied to MOBA games directly. First, they consume massive computation resources to construct the league. They use 12 copies of models in which there exist 128 TPU cores. Second, the scenarios between MOBA and RTS games are different. They leverage the nature of StarCraft II and use building orders to represent each strategy, which is unavailable in MOBA games.

In this paper, we propose a novel learning paradigm named **M**acro-**G**oals **G**uided (MGG) learning to train diverse policies in MOBA games. Motivated by top e-sports players, we define related game information as macro-goals to depict strategic decisions, which constitute macro-strategies in MOBA games. To explore macro-strategies efficiently, we extract macro-goals from human demonstrations and train a supervised model to predict macro-goals according to the given lineups and game information. We derive diverse policies by sampling macro-goals from the model, and incorporating the goals and an intrinsic reward into the original actor-critic framework to guide the policy learning process. We use the famous mobile 5v5 MOBA game, *Honor of Kings*[3], as our testbed, which is widely used for MOBA-game AI research Ye *et al.* [2020c,a,b]; Wu [2019]; Chen *et al.* [2021]; Wei *et al.* [2021]. Experimental results demonstrate that the AI agent we obtained can adopt different policies according to the given lineups and improve the Elo score by modelling policy diversities. Our contribution can be concluded as follows:

- To the best of our knowledge, we are the first to investigate the problem of policy diversity in MOBA games. We propose a novel framework that can learn diverse policies to adapt to different matches and lineups.

- Based on macro-goals, our methods can explore diverse macro-strategies efficiently and meanwhile preserve high-level micro-operations. Unlike league-based methods, our MGG framework uses the same computation resources as the original actor-critic framework.

- We conduct performance testing and diversity analysis on the full hero pool, i.e., 102 heroes. Experimental results show that our agents can outperform existing state-of-the-art MOBA-game AI systems and demonstrate diverse policies.

## 2 Related Works

### 2.1 Artificial Intelligence in MOBA Games

The progress of Game AI in MOBA mainly focuses on Dota 2 and *Honor of Kings*. OpenAI *et al.* [2019] have successfully defeated top e-sports players on playing 1v1 and 5v5 games in Dota 2 within limited heroes. Ye *et al.* [2020b,c,a] have proposed a series of works that exceed human-level performance and expanded hero pools in *Honor of Kings*. Besides, TencentHMS Wu [2019] exploits a hierarchical policy structure to control macro-strategy decisions and micro-level executions,

---

[2]https://www.reddit.com/r/MachineLearning/comments/bfq8v9/
[3]https://en.wikipedia.org/wiki/Honor_of_Kings

respectively. All the mentioned works except the supervised learning adopt a single strategy regardless of the heroes and lineups, which cannot take full advantage of heroes and lineups' characteristics.

## 2.2 Diverse Policies in Other Games

The league-based methods Vinyals *et al.* [2019]; Balduzzi *et al.* [2019]; Jaderberg *et al.* [2017, 2019] train multiple agents simultaneously to explore strategy space and require massive computation resources. However, it is impractical to use league-based methods in MOBA games. Because the space of strategies is complex in that there are $213,610,453,056(C_{40}^{10} \times C_{10}^5)$ lineups for 40 heroes in MOBA games and each lineup has various strategies. Therefore, it is difficult to design artificial signal to represent macro-strategies in MOBA games like Starcraft II (e.g., building order).

The aforementioned Game AIs design methods suffer from the following limitations: 1) the need for data on human behaviors; 2) heavy dependence of designer expert knowledge and substantial labor costs in searching a desirable behavior. Shen *et al.* [2020] can automatically generate diverse styles with almost no domain knowledge. However, this method is not suitable for complex games that last up to an hour long and consist of thousands of actions, such as MOBA games and RTS games.

## 2.3 Imitation Learning

In order to reduce the cost of strategy exploration in enormous lineups, we imitate macro-strategies from human demonstrations. Actually some works have trained reward functions from human preferences Nair *et al.* [2018]; Ibarz *et al.* [2018]; Christiano *et al.* [2017] or learnt policy directly from limited human data Ho and Ermon [2016]. Different from the above works, we abstract strategy as macro-goals in MOBA games. Our method not only imitates macro-strategies, but also leaves the agent to explore the optimal micro-operations (e.g., move, attack) by itself through external rewards.

## 2.4 Goal-based Learning

Goal-based methods Schaul *et al.* [2015]; Andrychowicz *et al.* [2017]; Zhu *et al.* [2021] are originally proposed to solve the problem of training a single goal-conditioned policy to accomplish multiple tasks. Some methods refer to the idea of goal-based RL and divide a task into a series of sub-tasks (i.e., goal) to accelerate the exploration and training of policy, such as automatic goal generation Florensa *et al.* [2018] and hierarchical RL Kulkarni *et al.* [2016]; Nachum *et al.* [2018], etc. However, in complex scenarios such as MOBA or RTS games, it is difficult to explore the meaningful goals, whether using GAN or training a high-level policy by RL methods. Therefore, we utilize the top players' demonstrations to pre-train a macro-level policy to generate reasonable and diverse macro-goals.

# 3 Macro-Goals Guided Reinforcement Learning

The problem we focused on can be formulated as a Markov Decision Process (MDP), defined by a tuple $(S, G, A, R, T, \gamma)$, where $S$ is the state space, $A$ is the action space, $T : S \times A \to S$ is the transition function, $R : S \times A \to R$ is the reward function, and $\gamma \in [0, 1]$ is the discount factor. In a typical MOBA game Silva and Chaimowicz [2017], the agents perform actions $a \in A$ (e.g. move, attack) according to the observations $s \in S$ (e.g. unit states, in-game stats) to achieve the final objective (e.g. destroy the crystal). In our setting, $G$ represents the macro-strategic state space in the game. And we define a macro-state extraction function $f : S \to G$ which maps the current state $s$ to the multi-dimensional macro-state $c = f(s)$. The macro-state $c \in G$ mainly includes the region state (e.g. the area on the map) and resource state (e.g. unit resource, gold, level). The macro-goal $g \in G$ represents the macro-state that the agent should achieve in the near future in a certain strategy.

Ye *et al.* [2020a] design the general reward function for MOBA games, including farming, KDA, damage, pushing, and win/lose related rewards. As a result of the single value system, the agent only learns to execute a single strategy from self-play training, that is, farming first, then fighting and pushing, no matter what strategy the opponents adopt. Thus, its macro-state set $\{c|c \in G_{RL}\}$ is only a part of $G$, i.e. $G_{RL} \subset G$. On the contrary, human strategies are diverse and their macro-state set $G_{human}$ can be considered to be close to the oracle $G$, i.e. $G_{human} = G_1 \cup \ldots G_i \cdots \cup G_N \approx G$, where $N$ is the number of human strategies. We define that strategy $i$ can be expressed as a set of macro-goal $\{g|g \in G_i\}$, where $G_i \subset G$. Our objective is to learn diverse strategies, that is, maximize

the macro-state entropy $-\sum_{c \in G} P(c)logP(c)$. Therefore, we model the policy as $\pi(a|s, g)$ and utilize $g \in G_{human} \approx G$ as the macro-goal of the policy. We also introduce an intrinsic reward function $R^{goal} : S \times G \times A \to R$ to guide its macro-state set $G_{RL}$ to be close to the complete $G$.

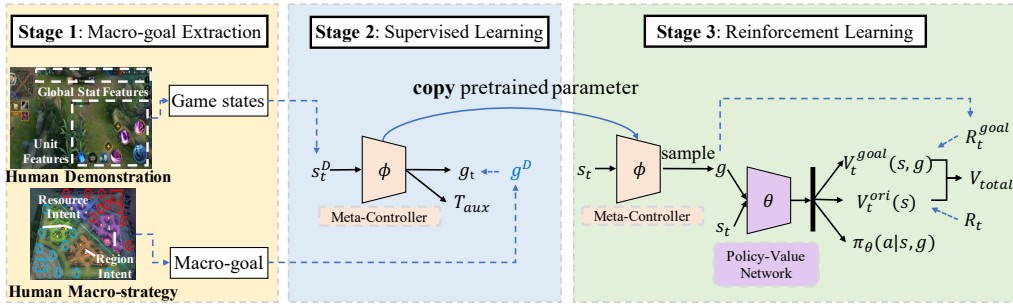

Figure 1: The MGG framework.

In this paper, we propose the **M**acro-**G**oals **G**uided (MGG) framework to learn diverse policies in MOBA games. Intuitively, top players' matches are observed to have high-quality and diverse macro-strategies but noisy micro-operations. Thus, completely imitating human demonstrations, including micro-operations, will worsen the performance of the policy. However, the macro-strategies from top players' matches can be used to imitate, which can reduce the cost of exploration in training. Our motivation is to define macro-goals to depict human strategies and then learn diverse strategies from these macro-goals only. In general, the MGG consists of three stages, as shown in Figure 1.

Stage 1: Extract state $s_t^D$ and its corresponding macro-goal $g^D$ from top e-sports demonstrations.

Stage 2: Use the extracted state and macro-goal pairs $\langle s_t^D, g^D \rangle$ as the training data to train the Meta-Controller parameterized by $\phi$, via supervised learning. During the training process, we use the auxiliary auto-encoding tasks $T_{aux}$ of lineups to improve prediction accuracy. Given a state $s$, the Meta-Controller $\phi$ should predict a macro-goal $g$ to depict the current strategy.

Stage 3: Use the Dual-clip PPO algorithm Ye *et al.* [2020a] to train the policy and value network $\theta$ conditioned on the macro-goal $g$. The macro-goals $g$ of the specified strategy is sampled from the pre-trained Meta-Controller. We introduce an intrinsic reward $R_t^{goal}$ to denote whether the agent achieves the macro-goal $g$. Besides, we use the multi-head value mechanism Ye *et al.* [2020a] to combine the original value output and the intrinsic value output.

To better illustrate the MGG method, we use *Honor of Kings* as a case study to describe some common terms associated with MOBA games. But it can be applied to other MOBA games (e.g. Dota 2 and *League of Legends*), as the playing mechanism across MOBA games are similar.

## 3.1 Generation of Macro-Goals

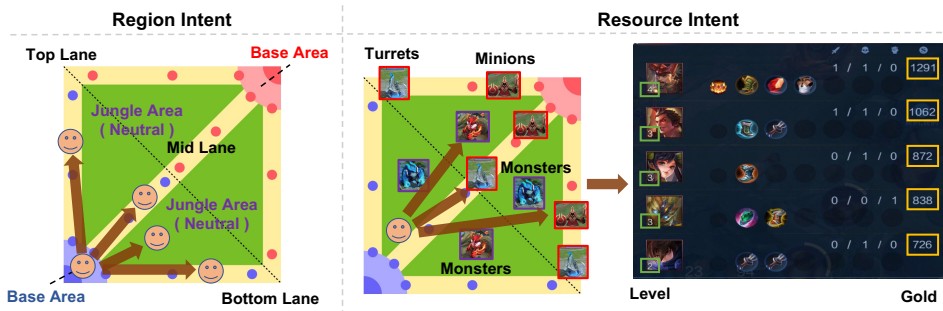

Figure 2: The macro-goal designed for abstracting player's macro-strategy. The red and blue on the map indicate two teams, and the map can be divided into 3 *lanes* (the top, middle, and bottom lane), 4 *jungle areas*, and 2 *base areas*. Circles represent *turrets*. The brown arrows in the left part indicate region intent ("Where to go"). The *turrets*, *monsters*, and *minions* on the right represent unit resources. The brown arrows on the right indicate resource intent ("What to do"), which means that the agent will kill units in order to level up or gain gold.

### 3.1.1 Macro-goal Definition

The motivation for designing MOBA AI macro-goals is inspired from how human players make strategic decisions. Human players usually make strategic decisions according to "Where to go" and "What to do". Thus, we define two types of macro-goals for all MOBA games, i.e., region intent ("Where to go") and resource intent ("What to do"), as shown in Figure 2. The region intent is the next region where the players move to, as shown in the left part of Figure 2. The resource intent is the quantity of game resources that players aim to obtain, including turrets, monsters, gold assignment, and other resources commonly found in MOBA games, as shown in the right part of Figure 2.

### 3.1.2 Macro-Goal Label Extraction

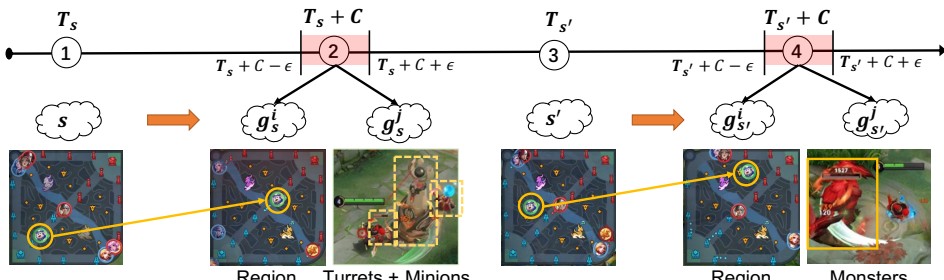

Figure 3: An example of macro-goal label extraction. The macro-goals at $T_s$ is to move into the opposite middle lane and gain gold resources from turrets and minions, while at $T_{s'}$ is to move to the upper jungle area and grab the monsters to level up. $C$ is the delta of frame numbers between the state and its extracted macro-goals.

We extract macro-goals from top e-sports players' matches. Given a state $s_t$ in the trajectory, we define a multi-dimensional goal and extract its region intents and resource intents in the future frame as label, i.e., macro-goal $g_t = f(s_{t+C})$. Each dimension denotes an aspect of the macro-goal in the current state. The multi-dimensional goal can guide our policy learning to achieve multiple goals at the same time. A concrete example of macro-goal label extraction is shown in Figure 3. The trajectory consists of frames from top players' matches. Let $T_s$ denotes the current frame. We sample the frame with goals from the interval $[T_s + C - \epsilon, T_s + C + \epsilon]$, where $C$ is a hyperparameter representing the delta of frame numbers between the state and its macro-goals, generally 30s. $\epsilon$ is a hyperparameter denoting the range of noise to avoid overfitting. Intuitively, by setting up labels in this way, we expect the Meta-Controller to learn the mapping from the current state $s_t$ to its future macro-goal $g_t$.

### 3.1.3 Meta-Controller Training

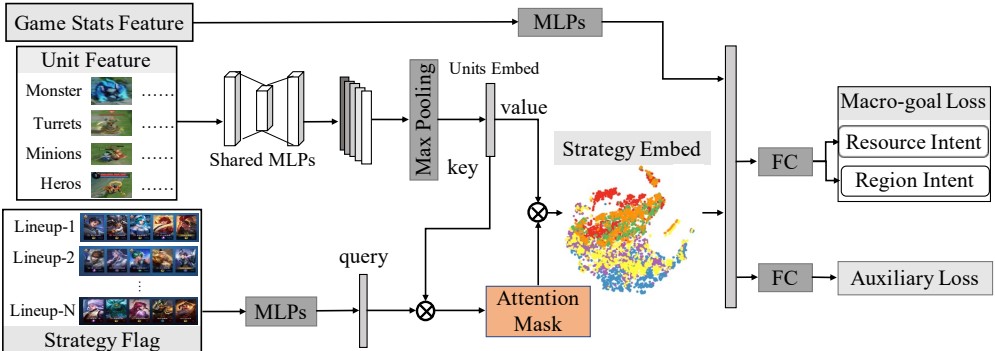

Figure 4: Network Architecture of the Meta-Controller. First, the features of each unit are extracted by some shared MLPs to the units embedding. Then, the lineup information is encoded by MLPs and used as a query vector to calculate the attention between the strategy and units embedding. Finally, the strategy embedding and game stats embedding are used to predict macro-goals $\phi(s_t)^g$ and strategy flag $\phi(s_t)^{aux}$.

After extraction of macro-goals, we train the prediction model, i.e., Meta-Controller, via supervised learning. The network structure of the Meta-Controller is shown in Figure 4. To predict macro-goals,

we input current game state features including observable unit's attributes and in-game statistics information, e.g., health point(hp), location, gold, level, etc. Note that our game state features include information of *heroes* and *lineups*, where the lineup means a combination of five heroes. Therefore, the Meta-Controller can predict macro-goals of different strategies according to the lineup features.

Beside, because the strategies in MOBA games have strong connections with lineups, we regard the auto-encoding task of lineups as an auxiliary task $T_{aux}$ to further model this relationship. It plays a role of learning latent representations of strategies. Then, we exploit the attention mechanism in which the latent representations of strategies as queries and the hidden encoding of game states as keys and values. In addition, to solve the imbalance of macro-goal labels and improve the

$$L^{SL}(\phi) = L_{FL}(\phi(s)^g, y^g) + \lambda * L_{CE}(\phi(s_t)^{aux}, y^{aux}), \tag{1}$$

where $L_{FL}$ is the focal loss of predicting labels, $L_{CE}$ is cross-entropy loss of the auxiliary task, and $\lambda$ is the balancing parameter. We use Adam Kingma and Ba [2014] to solve the optimization problem.

### 3.2 Macro-goals Guided Training Framework

We integrate the Meta-Controller into the original actor-critic framework as shown in Figure 5. Given the state $s_t$ at frame $t$, a macro-goal $g_t$ is generated by the pre-trained Meta-Controller. To guide the trained policy towards $g_t$, inspired by goal-based RL Andrychowicz *et al.* [2017]; Schaul *et al.* [2015], the policy and value function make decisions according to not only $s_t$ but also $g_t$. For each $N$ frames, the Meta-Controller updates a new goal $\hat{g}_t$ to follow the strategy. And the same goal $\hat{g}_t$ is kept during the whole $N$ frames $[t, t+N)$ to estimate parameters of policy $\pi_\theta(a|s_t, \hat{g}_t)$ and value functions $V_\theta(s_t, \hat{g}_t)$. Besides, we sample macro-goals from the softmax distribution of the Meta-Controller prediction to further increase the policy diversity even with the same lineups.

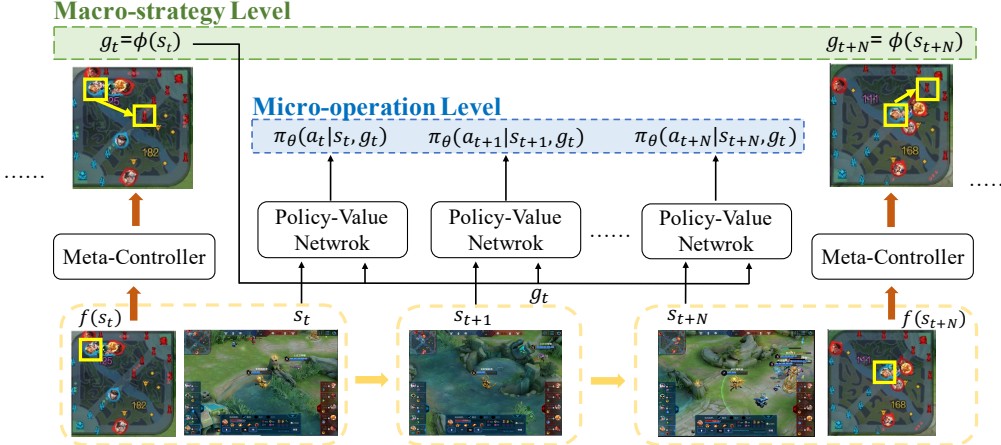

Figure 5: An example of a trajectory with MGG.

We design an intrinsic reward $R^{goal}$ to guide the policy towards the macro-goals as follows, which represents the delta of the distance between the current macro-state and the macro-goal:

$$R^{goal}(s_t, a_t, s_{t+1}, \hat{g}_t) = \left| f(s_t) - \hat{g}_t \right| - \left| f(s_{t+1}) - \hat{g}_t \right|, \tag{2}$$

where $\hat{g}_t = \phi(s_t)$ is the macro-goal generated by the Meta-Controller at frame $t$. $f$ is the macro-state extraction function, which is the same with the label extraction process. If the macro-state $f(s_{t+1})$ is more close to $\hat{g}_t$ than $f(s_t)$, the agent will receive a positive reward, and vice versa.

We combine the intrinsic reward $R^{goal}$ with the extrinsic environment reward, and the agent is trained to maximize the accumulated environment rewards as well as perform macro-goal distribution matching. Therefore, the agent can explore the best way to achieve the goal while learning the human strategy, so as to avoid the decline of micro-operation ability caused by completely imitation learning.

**Policy updates**. In order to avoid training instability in our large-scale distributed environment, we use the Dual-clip PPO method Ye *et al.* [2020a]. Differing from the original algorithm, we introduce the macro-goal $g$ into the policy $\pi_\theta(a_t|s_t, \hat{g}_t)$ and advantage estimation $\hat{A}_t(a_t, s_t, \hat{g}_t)$.

When $\pi_\theta(a_t|s_t, \hat{g}_t) \gg \pi_{\theta_{old}}(a_t|s_t, \hat{g}_t)$ and $\hat{A}_t < 0$, the radio $r_t(\theta) = \frac{\pi_\theta(a_t|s_t, \hat{g}_t)}{\pi_{\theta_{old}}(a_t|s_t, \hat{g}_t)}$ is huge, which causes the large and unbounded variance since $r_t(\theta)\hat{A}_t \ll 0$. Dual-clip PPO introduces another clipping parameter $c$ that indicates the lower bound when $\hat{A}_t < 0$. The new objective is the following:

$$L^{policy}(\theta) = \hat{\mathbb{E}}_t[\max(c\hat{A}_t, \min(clip(r_t(\theta), 1-\tau, 1+\tau)\hat{A}_t, r_t(\theta)\hat{A}_t))],$$

where $\tau$ is the original clip parameter in PPO.

**Value updates**. We use the multi-head value mechanism Ye *et al.* [2020a] to incorporate the value of our macro-goal. Specifically, the value of achieving macro-goals are regarded as one head. Therefore, the value loss is defined as follows:

$$L^{value}(\theta) = \hat{\mathbb{E}}_t[\sum_{head_k}(R_t^k - \hat{V}_t^k)], V_{total} = \sum_{head_k} w_k V_t^k(s_t, g_t),$$

where $w_k$ is the weight of the $k$-th head and $V_t^k(s_t, g_t)$ is the $k$-th value.

## 4 Experiments

In this section, we compare the state-of-the-art methods of supervised learning (SL) Ye *et al.* [2020b] and reinforcement learning (RL-baseline) Ye *et al.* [2020a] with the MGG method in the game of *Honor of Kings*, from the aspect of capability and diversity. Then, we compare MGG, RL-baseline and GAIL Ho and Ermon [2016] on some special lineup systems to evaluate the capability gains from increasing the policy diversity. Since the GAIL algorithm has a high sample complexity, which makes it is impossible to train on a full hero pool, i.e., 102 heroes. Thus we compare GAIL with other methods only on a small hero pool (20 heroes) in the case study section.

### 4.1 Experimental Settings

We construct a training dataset by collecting replays from the top 1% human players to train the Meta-Controller. We extract 4060-dimensional state and 64-dimensional macro-goal pairs $\langle s_t^D, g^D \rangle$ from the raw data, while the delta $C$ is 30 seconds and the noise $\epsilon$ is 3 seconds. We train only a single model for all heroes. We use 8 NVIDIA P40 GPUs for about 26 hours of training, and the batch size of each GPU is set to 512. We set $\alpha = 0.75, \gamma = 2$ for focal loss $L_{FL}$ Lin *et al.* [2017], and set $\lambda = 1$ for the weight of auxiliary task. We use Adam with the initial learning rate of 0.0001.

To make a fair comparison, except for the additional macro-goal features and intrinsic reward, the rest of the MGG network architecture and training settings are exactly the same as the RL-baseline. MGG and other RL methods adopt self-play training and train by randomly selecting heroes over a physical computer cluster with 60,000 CPUs and 830 NVIDIA V100 GPUS. The batch size of each GPU is set to 4096. All methods converged after training for a total of 420 hours. We evaluate all methods on 102 heroes[4] in Section 4.2 and 4.3 and on about 20 heroes in Section 4.4. For online evaluation, the response time of AI is set to 193ms, including observation delay (133 ms) and response delay (60 ms). Response delay consists of features, models, result processing, and transmission delays. The average APMs of AI and top players are comparable (80.5 and 80.3, respectively).

### 4.2 AI Performance

We compare MGG with built-in bots, supervised learning agents Ye *et al.* [2020b] (SL), and the original reinforcement learning agents Ye *et al.* [2020a] (RL-baseline). Note that RL-baseline is also the state-of-the-art method in the game of *Honor of Kings*. We use all methods to play pairwise matches, 400 games for each pair, and then calculate the Elo scores Coulom [2008]. All lineups in each match are randomly selected from the full 102 hero pool.

In addition, during the period from April 1st to April 4th, 2021, we conduct a large-scale human-machine test in the game activity to evaluate the average level of human players. We run about 100,000 tests on RL methods, where players are randomly matched against the RL-baseline and MGG agents without being told about the opponent model, just knowing that the opponent is the AI.

---

[4]see Supplementary for more details.

And we also conduct 50,000 tests on the SL method. The win rates of SL, RL-baseline, and MGG against humans are 93.10%, 97.77%, and 97.81%, respectively.

Figure 6 shows that the MGG method performs better than all the baseline methods in real games, demonstrating the benefits from modelling policy diversities and taking advantage of the characteristics of lineups.

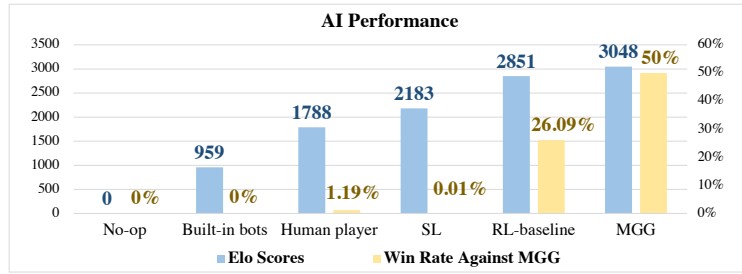

Figure 6: The performance comparison between MGG and the baseline methods in the game of *Honor of Kings*. **Blue:** Overall comparison of Elo scores. **Yellow:** Baselines vs. MGG win rate.

### 4.3 Diversity in Different Lineups and Matches

#### 4.3.1 Diversity in Different Lineups

We visualize macro-state embeddings of MGG, RL-baseline, and human players in Figure 7, and points are colored according to different methods. Figure 7 shows that MGG plays more diverse policies than RL-baseline. The macro-state space of MGG (blue points) covers both the RL-baseline (red points) and human (yellow points), which indicates MGG can play both the human strategies and RL-baseline strategies. Besides, we can observe that the points of MGG and human strategy are clearly separated from each other. Conversely, the points of the RL-baseline strategy are mixed.

We evaluate the Davies-Bouldin Index Davies and Bouldin [1979] of macro-states embeddings in different lineups, as shown in Table 1. We can see the MGG has a lower Davies-Bouldin Index, which indicates the MGG's macro-strategies have a better separation between different lineups.

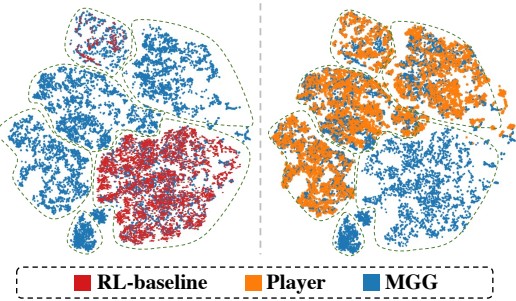

Table 1: Davies-Bouldin Index of MGG and RL-baseline in different lineups. The values closer to zero indicate a better partition, meaning that the strategies (macro-state space) varies more between the different lineups.

|  | Davies-Bouldin Index |
|---|---|
| RL-baseline | 10.72 |
| MGG | 1.76 |
| human | 1.45 |

Figure 7: **Macro-states t-SNE Embedding of different algorithms in the same space**. **Left:** comparison of MGG (blue points) and RL-baseline (red points). **Right:** comparison of MGG and human player (yellow points). The strategy space of MGG covers both.

#### 4.3.2 Diversity in Different Matches

The previous section shows that the MGG agent presents diverse policies in different lineups. To evaluate the diversity when the agents pick the same lineup in different matches, we define the macro-state entropy metric $H = -\sum_{c \in G} P(c) log P(c)$, where $P(c)$ is the probability mass function of macro-state and $c = f(s)$. In this experiment, we run 1000 matches for each comparison, and we let agents play against the same bot with the same lineup and start the game from the same state.

In Table 2, we compare MGG with the RL-baseline, which shows that the MGG agent plays diverse polices even in the same lineup due to sampling the macro-goal from the Meta-Controller in real-time.

Table 2: Comparing macro-state entropy of each method in the different matches while picking the same lineup.

| | Built-in bots | RL-baseline | MGG |
|---|---|---|---|
| macro-state entropy | 0.000 | 0.014 | **0.408** |

## 4.4 Case Study

In *Honor of Kings*, the most common strategy for human players is three-lane-strategy, in which the marksman, mage, and warrior (or tank) obtain resources from three lanes, respectively, the assassin grabs the neutral jungle resources, and the supporter roams to assist them without gaining resources. Therefore, the general reward function Ye *et al.* [2020a] is designed following the common strategy, so agents will perform similar strategies in different lineups and thus cannot perform the most suitable strategy in some special lineups. To evaluate the effectiveness of MGG on this issue, we select several lineup systems[5] that differed significantly from the common strategy to evaluate the diversity and capabilities of MGG and other methods.

We also compare MGG with GAIL Ho and Ermon [2016], which is shown to be effective in behavior imitating tasks. GAIL uses a discriminator to distinguish whether a state-action pair is from the human player or the learned policy, meanwhile optimizes the policy to confuse the discriminator. However, adversarial training will greatly affect the training efficiency of the policy, so we only train separately under certain special lineup systems, not the full hero pool.

### 4.4.1 Case of Marksman-Core System

*Marksman-Core System* refers to substituting supporter for the mage so that the powerful marksman can obtain the resources of the two lanes to gain an advantage over the enemy in money and equipment as fast as possible. Therefore, we evaluate the achievement of the resource allocation strategy within the team by analyzing the money rate of the marksman ($\frac{money_{marksman}}{\sum(money_{team\_index})}$) during the match against the RL-baseline, and then analyze the advantage over the enemy through the marksman's money difference ($money_{marksman} - \frac{1}{5}\sum(money_{enemy\_index})$). As shown in Figure 8, compared to RL-baseline and GAIL, MGG is closer to the distribution of human players in terms of marksman money rate and money difference, indicating that applying MGG is more beneficial for achieving the resource intent strategies, which is difficult for the original reward function to achieve.

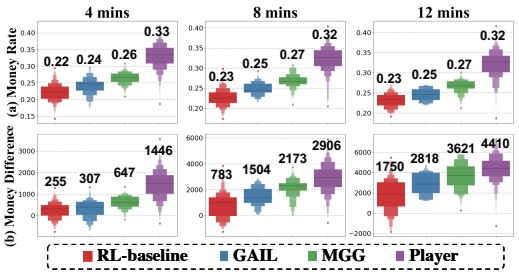

Figure 8: Statistical indicator of resource allocation in *Marksman-Core System*.

Table 3: Neutral monster resource metrics of *Resource-Grab System*. In the early stage, the near side resources are at most 2 and the far side resources are also capped at 2.

| | Near Side Resources | Far Side Resources | Total Resources |
|---|---|---|---|
| Human | 1.317 | 0.561 | 1.878 |
| RL-baseline | 1.002 | 0.000 | 1.002 |
| GAIL | 1.141 | 0.056 | 1.197 |
| MGG | **1.741** | **0.658** | **2.399** |

### 4.4.2 Case of Resource-Grab System

*Resource-Grab System* means using a strong assassin in the early stage (first 2 minutes) to grab the resources of neutral monsters closer to the opponents, which belongs to the long-term regional intent strategy. Therefore, the success of the strategy can be expressed by the number of neutral resources snatched in the early stage. As shown in Table 3, RL-baseline maximizes short-term gains by only grabbing the resources closest to it based on the common policy. In contrast, MGG effectively implements this long-term strategy and outperforms the human by 0.52 monster resources in total. Besides, GAIL is difficult to guide agents at the micro-operation level to implement the regional intent for grabbing the enemy's resources because this strategy requires long-term planning.

---

[5]see Supplementary for more cases of lineup systems and results of their human-machine test.

### 4.4.3 Capability Analysis

To further investigate whether increased diversity can improve the capability of the particular lineup systems, we test the methods against each other, with pairs playing 100 games. In Figure 9, the Elo scores of built-in bots, SL agents, the RL-baseline agents, GAIL agents, and MGG agents are provided. Elo scores demonstrate that different strategies have a significant impact on the upper limit of its capability for the same lineup. Figure 9 shows that with improved strategy diversity, MGG increases 312 and 560 Elo scores over baseline in the above two lineup systems, respectively. Thus, training MOBA AI with strategies diversity and enhancing specific lineup systems' capabilities benefits significantly from our proposed method. The video of the above case can be found in https://sites.google.com/view/mgg-demo.

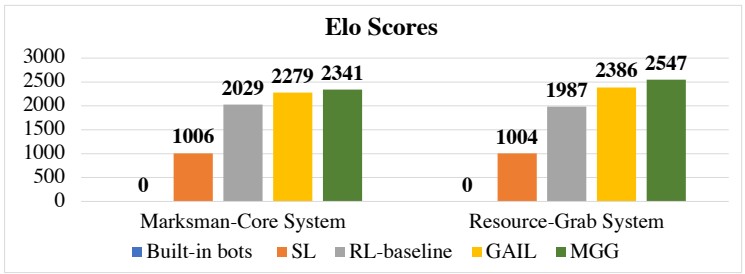

Figure 9: Comparison of Elo scores between MGG and other methods in two special lineup systems.

## 5 Conclusions

In this paper, we propose the MGG framework to learn diverse policies during training. We define macro-goals to depict diverse strategies in MOBA games. We sample macro-goals from softmax distribution of the Meta-Controller prediction, which enhances policy diversity. To guide the policy towards a specific strategy, we design an intrinsic reward to indicate whether the agents achieve macro-goals during training. The trained agents can play diverse policies in different matches and exploit lineups' characteristics to improve capabilities. In the future, after learning diverse policies, we will investigate non-transitivity in strategies of MOBA games using game theory. We will also try to find a solution that is closer to Nash equilibrium in MOBA games.

## 6 Broader Impact

**To the research community.** MOBA (Multiplayer Online Battle Arena) poses a grand challenge to the AI community. Even though the existing MOBA-game AI systems have achieved or even exceeded human-level performance, they still suffer from the lack of policy diversity in such a complex environment. To this end, this paper is introducing a learning paradigm to train diverse policies for the MOBA-game AI systems. We herewith expect that this work can provide inspiration for the diversity of strategies in various game AI research.

**To the game industry.** Our AI has found several real-world applications in the game, and is changing the way that MOBA game designers' work, elaborated as follows: 1) Game balance testing. In MOBA and many other game types, balancing the ability of each strategy is essential. Using similar techniques presented in this paper is an easy way to construct a strategy balance testing tool for MOBA games. 2) PVE (player vs environment) game mode. Introducing AI with diverse strategies into the game's PVE mode is a low-cost method to increase the interest of players.

**To the e-sports community.** Like AlphaGo, our method can provide a high-quality low-cost training environment with diverse strategies for e-sports players. The AI system with diverse macro-strategies and high-level micro-operations is suitable as a training partner for e-sports athletes.

## 7 Funding Disclosure

This work is supported by the Tencent AI Department and the Tencent TiMi Studio.

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
