# A Appendix

## A.1 Additional Case

In addition to *Marksman-Core System* and *Resource-Grab System* mentioned in the paper, we also make a case study on *Teleportation System*.

### A.1.1 Case of Teleportation System

*Teleportation System* is an exceptional strategy. The key is Da Qiao, a support hero with 2 teleport skills. The Skill I can send teammates back to the spring, and Skill II can teleport teammates to Da Qiao's vicinity. Players use these teleport skills to achieve regional intent, such as rapid and long-distance support and timely rescue. We use the teleport ratio to evaluate the regional intention's achievement; that is, the number of teammates teleported each time the skill is released. In Table 1, both skills' teleport rates are much lower than that of human players, indicating that agents basically have no macro intention of long-distance transfer in the common strategy. We observe that MGG learns this strategy and makes full use of this strategy for scheduling. Compared with human players, Skill I and Skill II's teleport rates increase by 0.76 and 0.92, respectively. Specifically, we also find that although GAIL can guide agents to learn more teleportation from the micro-operation level, it still cannot elevate this strategy to the human level.

Table 1: Comparing teleport rates of each algorithm in the *Teleportation System*.

| Method | Skill I | Skill II |
|---|---|---|
| Player | 0.70 | 1.73 |
| RL-baseline | 0.08 | 0.40 |
| GAIL | 0.14 | 0.84 |
| MGG | **1.46** | **2.65** |

### A.1.2 Extra Experiment of Resource-Grab System

In addition to the success of the strategies presented in the paper, we also test the capabilities of the different strategies in the this lineup system. Since the RL-baseline using common strategy does not know how to deal with this strategy, we use agents of RL-baseline and MGG to test the advantages against the invited high-ranking players in each of the 30 matches on September 10th, 2020. The result is listed in Table 2, where we analyze the match statistics metrics of the powerful assassin at the end of the game. Although the RL-baseline picks the same *Resource-Grab System* lineups, the common strategy still limits the maximum strength. As the results show, MGG can better learn the human special strategy and surpass human players in indicators such as experience, money, kills, and assistance, proving its effectiveness.

Table 2: Match statistics of *Resource-Grab System*. 30 games are human players vs. human players. Baseline and MGG agents also each play 30 games against human players.

| Method | Experience | Money | Damage | **K**ill/**D**eath/**A**ssist |
|---|---|---|---|---|
| Player | 14573.92 | 10612.11 | 86841.07 | 9.6/2.7/8.2 |
| RL-baseline | 13110.25 | 9819.34 | 64016.12 | 7.0/2.1/8.2 |
| MGG | **14659.71** | **10839.80** | 86242.77 | **10.2**/2.7/**10.3** |

### A.1.3 Extra Experiment of Marksman-Core System

We also invite professional esports players of *Honor of Kings* to play four matches against our AI and baseline AI on July 3rd, 2020. In all four matches, AI chooses the lineups of *Marksman-Core System* against the players' regular lineups.

Table 3 shows the match results. We see that MGG can defeat the same human team faster and complete the game with fewer deaths. Due to confidentiality agreements, we can't reveal any more details about the matchs.

The core of the system is that teammates give more resources to the marksman in the early stage to quickly open the money gap with opponents. The five then gather to quickly defeat defenders and turrets in one lane for an early victory. This is exactly the strategy used by MGG in both games. Instead, RL-Baseline uses the common strategy of MOBA games, with each role fighting on their own lane and steadily destroying the three-lane turrets. Thus, dying less or finishing the game early proves that a sound strategic choice can take full advantage of the lineups, as MGG has done.

Table 3: Results of MGG/RL-baseline vs. Top Human Players

|  | score | kill | death | kill/death | game length |
| --- | --- | --- | --- | --- | --- |
| RL-baseline | 2:0 | 29.7 | 11.3 | 2.7 | 14'52" |
| MGG | 2:0 | **27** | **2** | **17** | **10'9'** |

We also invite professional esports players of *Honor of Kings* to play against our AI and baseline AI on July 3rd, 2020. Table 3 shows the match results. We see that MGG can defeat the same human team faster and complete the game with fewer deaths.

## A.2 Infrastructure Design

In Figure 1, we show our infrastructure. It consists of four major components: AI Server, Inference Server, RL Learner and Memory Pool. The AI Server (the Actor) covers the interaction logic between the agents and environment. The Inference Server is for centralized batch inference on the GPU side. The RL Learner (the Learner) is a distributed training environment for RL model training. And the Memory Pool is for storing experience replay, implemented as a memory-efficient circular queue.

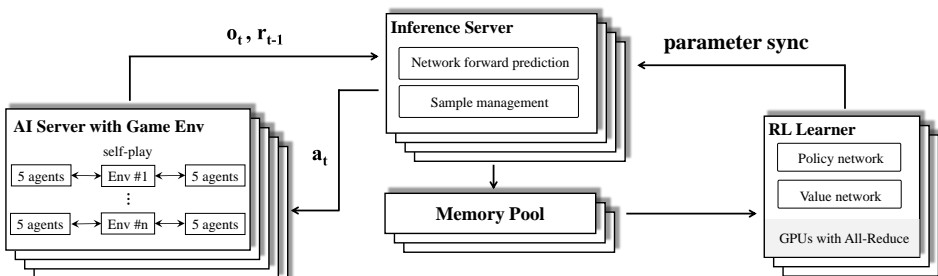

Figure 1: Our infrastructure design.

We used a large amount of computing resources for building our AI, due to the complex nature of the problem we study. In fact, the computing resources required for complex game-playing AI programs are non-trivial, e.g., AlphaGo Lee Sedol version (280 GPUs), OpenAI Five Final (1920 GPUs), and the final version of AlphaStar (3072 TPUv3 cores). We will continue to work on the infrastructure efficiency to further reduce the computational cost.

## A.3 Game Environment

In Figure. 2, we show a game UI of *Honor of Kings*. All the experiments in the paper were carried out using a fixed big version (Version 1.54 series) of game core of *Honor of Kings* for fair comparison.

## A.4 Hero Pool

The heroes used for constructing the lineups are summarized in Table 4. The first row of the Table4 represents all heroes we used for testing performance and diversity. The next three rows of the table represent the heroes we used for the case study, with the key hero bolded in each strategy .

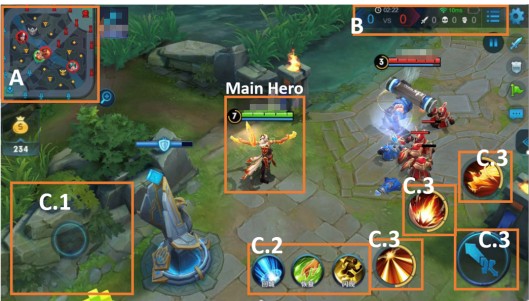

Figure 2: Game UI of *Honor of Kings*. The hero controlled by the player is called "main hero". Bottom-left is the movement controller (C.1), while the right-bottom set of buttons are ability controllers (C.2, C.3). Players can observe game situations via the screen (local view), obtain game states with dashboard (B), and obtain a global view with the top-left mini map (A).

Table 4: Heroes constructing the lineups.

| Pool | Randomly pick 5 heroes to form a lineup. |
|---|---|
| All Hero-102 | Lian Po, Xiao Qiao, Zhao Yun, Mo Zi, Da Ji, Ying Zheng, Sun Shangxiang, Luban Qihao, Zhuang Zhou, Liu Chan Gao Jianli, A Ke, Zhong Wuyan, Sun Bin, Bian Que, Bai Qi, Mi Yue, Lv Bu, Zhou Yu, Yuan Ge Xia Houdun, Zhen Ji, Cao Cao, Dian Wei, Gongben Wucang, Li Bai, Make Boluo, Di Renjie, Da Mo, Xiang Yu Wu Zetian, Si Mayi, Lao Fuzi, Guan Yu, Diao Chan, An Qila, Cheng Yaojin, Lu Na, Jiang Ziya, Liu Bang Han Xin, Wang Zhaojun, Lan Lingwang, Hua Mulan, Ai Lin, Zhang Liang, Buzhi Huowu, Nake Lulu, Ju Youjing, Ya Se Sun Wukong, Niu Mo, Hou Yi, Liu Bei, Zhang Fei, Li Yuanfang, Yu Ji, Zhong Kui, Yang Yuhuan, Chengji Sihan Yang Jian, Nv Wa, Ne Zha, Ganjiang Moye, Ya Dianna, Cai Wenji, Taiyi Zhenren, Donghuang Taiyi, Gui Guzi, Zhu Geliang Da Qiao, Huang Zhong, Kai, Su Lie, Baili Xuance, Baili Shouyue, Yi Xing, Meng Qi, Gong Sunli, Shen Mengxi Ming Shiyin, Pei Qinhu, Kuang Tie, Mi Laidi, Yao, Yun Zhongjun, Li Xin, Jia Luo, Dun Shan, Sun Ce Zhu Bajie, Shangguan Waner, Ma Chao, Dong Fangyao, Xi Shi, Meng Ya, Luban Dashi, Pan Gu, Chang E, Meng Tian Jing, A Guduo |
| Marksman-Core System-9 | **Hou Yi, Sun Shangxiang** Liu Bang, Zhao Yun, Gongben Wuzang, Cai Wenji, Zhuang Zhou, Ming Shiyin, Taiyi Zhenren |
| Resource-Grab System-13 | **Pei Qinhu, Baili Xuance** Lv Bu, Hua Mulan, Baiqi, Zhang Liang, Wang Zhaojun, Yang Yuhuan, Dong Fangyao, Lao Fuzi Zhang Fei, Sun Bin, Niu Mo |
| Teleportation System-11 | **Da Qiao** Lao Fuzi, Dong Fangyao, Zhao Yun, Nake Lulu, Shen Mengxi, Buzhi Huowu, Diao Chan, Make Boluo, Hou Yi, Li YuanFang Zhang Fei, Sun Bin, Niu Mo |

### A.5 Network Architecture of Reinforcement Learning

The policy-value network is to predict the actions of the current game states. The input features include both scalar and spatial features. Scalar features are made up of observable units' attributes, game statistics information. Spatial features are extracted from the player's local view map. Besides, there is a macro-goal feature that is sampled from the Softmax distribution of meta-controller. It plays a role in guiding the training of policies towards the macro-goal.

The observations are processed by a deep LSTM, which maintains memory between steps. To improve the prediction effects of the model, we use target attention to help target selection, and we design action mask to eliminate unnecessary RL explorations.

We develop hierarchical actions head, including three parts: 1) what action to take; 2) who to target; 3) how to act. The overall architecture of our policy-value network is shown in Figure 3.

### A.6 Agent Action

Table 5 provides the details of our action space design.

### A.7 Reward Design

Table 6 shows the details of our reward.

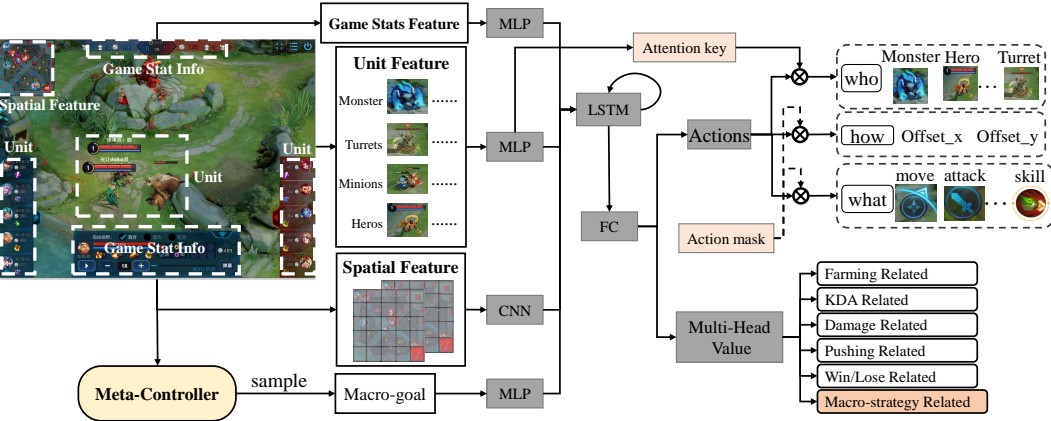

Figure 3: Policy-value Neural Network.

Table 5: Agent action space.

| Action | Detail | Description |
|---|---|---|
| What | Illegal action | Placeholder. |
| | None action | Executing nothing or stopping continuous action. |
| | Move | Moving to a certain direction determined by move x and move y. |
| | Normal Attack | Executing normal attack to an enemy unit. |
| | Skill1 | Executing the first skill. |
| | Skill2 | Executing the second skill. |
| | Skill3 | Executing the third skill. |
| | Skill4 | Executing the fourth skill (only a few heroes have Skill4). |
| | Summoner ability | An additional skill choosing before the game begins (10 to choose). |
| | Return home(Recall) | Returning to spring, should be continuously executed. |
| | Item skill | Some items can enable an additional skill to player's hero. |
| | Restore | Blood recovering continuously in 10s, can be disturbed. |
| | Collaborative skill | Skill given by special ally heroes. |
| How | Move X | The x-axis offset of moving direction. |
| | Move Y | The y-axis offset of moving direction. |
| | Skill X | The x-axis offset of a skill. |
| | Skill Y | The y-axis offset of a skill. |
| Who | Target unit | The game unit(s) chosen to attack. |

## A.8 Feature Design

The detailed features extracted by policy-value network are listed in Table 7. The feature consists of two main types: scalar features and spatial features. Scalar features include unit attributes, in-game statistics and invisible opponent information. Note that the invisible opponent information is only applied to the value network during training.

The way of feature normalization is as follows. For continuous features, we use their maximum and minimum values to normalize them into the interval of [0,1], such as health point (HP), mana, speed, etc. For example, the HP of a hero is normalized to a value between 0 (death) and 1 (full health). And for discrete features, we use one-hot representation by enumerating all possible values, such as skill level, kill-death-assist stats, etc. For example, the in-game skill level of a hero can be level 1 to level 15, so we use a one-hot vector of dimension 15 for representation.

The detailed features of meta-controller are a subset of the above features, as shown in Figure 8. The meta-controller network only needs scalar features, including unit attributes and in-game statistics information. Besides, we input the macro-strategy flag features including type of macro-strategies and lineups, to help the meta-controller learn latent representations of strategies.

Table 6: Reward design details.

| Head | Reward Item | Weight | Type | Description |
|---|---|---|---|---|
| Farming Related | Gold | 0.005 | Dense | The gold gained. |
| | Experience | 0.001 | Dense | The experience gained. |
| | Mana | 0.05 | Dense | The rate of mana (to the fourth power). |
| | No-op | -0.00001 | Dense | Stop and do nothing. |
| | Attack monster | 0.1 | Sparse | Attack monster. |
| KDA Related | Kill | 1 | Sparse | Kill a enemy hero. |
| | Death | -1 | Sparse | Being killed. |
| | Assist | 1 | Sparse | Assists. |
| | Tyrant buff | 1 | Sparse | Get buff of killing tyrant, dark tyrant, storm tyrant. |
| | Overlord buff | 1.5 | Sparse | Get buff of killing the overlord. |
| | Expose invisible enemy | 0.3 | Sparse | Get visions of enemy heroes. |
| | Last hit | 0.2 | Sparse | Last hitting an enemy minion. |
| Damage Related | Health point | 3 | Dense | The health point of the hero (to the fourth power). |
| | Hurt to hero | 0.3 | Sparse | Attack enemy heroes. |
| Pushing Related | Attack turrets | 1 | Sparse | Attack turrets. |
| | Attack crystal | 1 | Sparse | Attack enemy home base. |
| Win/Lose Related | Destroy home base | 2.5 | Sparse | Destroy enemy home base. |
| **Macro-strategy Related** | Macro-strategy | 1.0 | Sparse | The distance between the current state and macro-goal. |

Table 7: Feature details of Policy-Value Network.

| Feature Class | Field | Description | Dimension |
|---|---|---|---|
| **1. Unit feature** | Scalar | Includes heroes, minions, monsters, and turrets | 8599 |
| Heroes | Status | Current HP, mana, speed, level, gold, KDA, buff, bad states, orientation, visibility, etc. | 1842 |
| | Position | Current 2D coordinates | 20 |
| | Attribute | Is main hero or not, hero ID, camp (team), job, physical attack and defense, magical attack and defense, etc. | 1330 |
| | Skills | Skill 1 to Skill N's cool down time, usability, level, range, buff effects, bad effects, etc. | 2095 |
| | Item | Current item lists | 60 |
| Minions | Status | Current HP, speed, visibility, killing income, etc. | 1160 |
| | Position | Current 2D coordinates | 80 |
| | Attribute | Camp (team) | 80 |
| | Type | Type of minions (melee creep, ranged creep, siege creep, super creep, etc.) | 200 |
| Monsters | Status | Current HP, speed, visibility, killing income, etc. | 868 |
| | Position | Current 2D coordinates | 56 |
| | Type | Type of monsters (normal, blue, red, tyrant, overlord, etc.) | 168 |
| Turrets | Status | Current HP, locked targets, attack speed, etc. | 520 |
| | Position | Current 2D coordinates | 40 |
| | Type | Type of turrets (tower, high tower, crystal, etc.) | 80 |
| **2. In-game stats feature** | Scalar | Real-time statistics of the game | 68 |
| Static statistics | Time | Current game time | 5 |
| | Gold | Golds of two camps | 12 |
| | Alive heroes | Number of alive heroes of two camps | 10 |
| | Kill | Kill number of each camp | 6 |
| | Alive turrets | Number of alive turrets of two camps | 8 |
| Comparative statistics | Gold diff | Gold difference between two camps | 5 |
| | Alive heroes diff | Alive heroes difference between two camps | 11 |
| | Kill diff | Kill difference between two camps | 5 |
| | Alive turrets diff | Alive turrets difference between two camps | 6 |
| **3. Invisible opponent information** | Scalar | Invisible information used for the value net | 560 |
| Opponent heroes | Position | Current 2D coordinates, distances, etc. | 120 |
| NPC | Position | Current 2D coordinates of all non-player characters, including minions, monsters, and turrets | 440 |
| **4. Spatial feature** | Spatial | 2D image-like, extracted in channels for convolution | 6x17x17 |
| Skills | Region | Potential damage regions of ally and enemy skills | 2x17x17 |
| | Bullet | Bullets of ally and enemy skills | 2x17x17 |
| Obstacles | Region | Forbidden region for heroes to move | 1x17x17 |
| Bushes | Region | Bush region for heroes to hide | 1x17x17 |

Table 8: Feature details of Meta-controller.

| Feature Class | Field | Description | Dimension |
|---|---|---|---|
| **1. Unit feature** | Scalar | Includes heroes, minions, monsters, and turrets | 3946 |
| Heroes | Status | Current HP, mana, speed, level, gold, KDA, and magical attack and defense, etc. | 1562 |
| | Position | Current 2D coordinates | 20 |
| Minions | Status | Current HP, speed, visibility, killing income, etc. | 920 |
| | Position | Current 2D coordinates | 80 |
| Monsters | Status | Current HP, speed, visibility, killing income, etc. | 728 |
| | Position | Current 2D coordinates | 56 |
| Turrets | Status | Current HP, locked targets, attack speed, etc. | 540 |
| | Position | Current 2D coordinates | 40 |
| **2. In-game stats feature** | Scalar | Real-time statistics of the game | 104 |
| Static statistics | Time | Current game time | 57 |
| | Camp | types two camps | 1 |
| | Alive heroes | Number of alive heroes of two camps | 10 |
| | Kill | Kill number of each camp | 6 |
| | Alive turrets | Number of alive turrets of two camps | 8 |
| Comparative statistics | Alive heroes diff | Alive heroes difference between two camps | 11 |
| | Kill diff | Kill difference between two camps | 5 |
| | Alive turrets diff | Alive turrets difference between two camps | 6 |
| **3. Strategy-flag feature** | Scalar | Typs of strategies and lineups | 10 |