# OpenReview forum: "Learning Diverse Policies in MOBA Games via Macro-Goals"
_NeurIPS.cc/2021/Conference — NeurIPS 2021 Poster_

### Official Review · Reviewer_y4xZ · 2021-07-04

**Rating:** 7
**Confidence:** 4

**Summary:**

The paper addresses RL agent training in MOBA games, specifically the ability to acquire a diverse set of policies when playing a variety of strategic scenarios (team compositions, aka lineups). Human example data is processes to extract future states reached from initial states, treating these future states as goals. A supervised learning model is trained to predict the future goal state from a given game state using an attention module that accounts for the strategic scenario (lineup) and game state (unit features & aggregate game stats). An existing RL framework is then augmented to condition on this goal state and trained with a reward that encourages reducing the distance to this goal in addition to the baseline reward from the environment. A wide variety of results show the method is better able to adapt to unique strategic scenarios (lineups) and exhibits a broader array of behaviors compared to the baseline agent or an imitation learning agent (GAIL) while retaining strong performance against human players.


**Ethical Concerns:**

The paper does not address how consent was obtained from players, only that players were informed of being in the experiment. Given the online game setting players may not have had a choice - it's not clear whether/how this fits into standard ethics review protocols (IRB & associated documentation). I'm not confident this crosses any ethical lines, but I would like to flag the topic for expert review.

Note that the checklist indicates "We declare the test instructions in the game", but these are not included in the paper / supplement, so I have no way to review them.
Similarly the checklist answer about hourly wages indicates "We give out in-game rewards for testing", but what these are and how they should be valued is not indicated. Again, a tricky issue as game rewards are hard to convert to real world value/currency, but I wanted to hedge toward being safe.

I would also note that NeurIPS may want to establish some specific details on ethics policy for running experiments in live services (including games) - many companies operate live services where things like IRB documentation, informed consent, and compensation are all different to standard academic procedure. To the best of my knowledge these A/B tests are often conducted without informed consent (often without information). Whether running A/B tests of user interface design or playmate AI, these types of experiments stike me as topics that will appear across many papers appropriate to NeurIPS.

**Ethics Review Area:**

["Privacy and Security (e.g., consent)", "Responsible Research Practice (e.g., IRB, documentation, research ethics)"]

**Limitations And Societal Impact:**

No. Potential negative societal impacts are not discussed. MOBAs as real-time control problems feel like natural models for military automation tasks (training agents to perform various team activities). The paper-specific design to allow specification of a goal that controls agent behavior is easily applied to controlling RL agents toward whatever an actor using the agent may desire, including potentially harmful actions.

No limitations of what the method can achieve is discussed. These are explained above under "Quality". For example, the Figure 8 results show a substantial gap remaining between the MGG method and player behavior - why?. In Figure 7 MGG produces large clusters of behaviors human players do not - what might this indicate? In Figure 6b the differences between MGG and RL-baseline around on the order of 0.04% - is this statistically significant? Why is it so small? None of these are major failings, but they are not framed as limitations to be considered or explained.

**Main Review:**

The paper has several strengths:
- Strong empirical results.
- Variety of experimental comparisons. The experiments test a variety of features of the approach in terms of diversity of strategies, adaptability to new lineups, and overall competitive performance.

The paper's primary limitations:
- Limited novelty: Using imitation learning to bootstrap long horizon decisions from human data is not novel per se. The novelty lies in the combination of elements.
- Lacking discussion of limitations and remaining open problems. As detailed below there are some areas where there is a gap between the performance of the method and the ideal that should be discussed.


Below are comments specific to the review criteria prompts.

# Originality

Using imitation learning to bootstrap long horizon decisions from human data is not novel, nor is goal-conditioned learning. The primary technical novelties here lie in the specific modeling choices and their integration. I believe this integration is novel and valuable given the complexity of the domain and problem. It is clear this builds on prior efforts while enabling diversification of policies in a different way.

I am not sure if the attention mechanism employed for learning goals to predict is specifically new, but it seems to be.

Related work is adequately cited for the relevant techniques to compare.

# Quality

The results partially support the claims. Some specific points:
1. The new method does not seem to improve over the baseline agent against humans: Figure 6b shows a ~0.04% difference in (cumulative) win rate. The text claims this is superior performance but it does not seem obvious that is true (and there is no statistical test as the evaluation was performed once).
2. MGG does cover more of behavior space compared to the RL-baseline or players (Figure 7). But MGG also more parts of the space that players do not occupy. It is not clear whether this is a good thing or a problem.
3. There is no comparison of computational costs of the techniques. While greater efficiency is not claimed per se, this would help complete the picture.
4. Figure 8 results show a substantial gap remaining between the MGG method and player behavior.

The methods employed are appropriate for training RL agents in team games, and incorporating goal prediction into the policy training is clear incremental improvement over prior work.

The paper is not careful to address weaknesses or remaining gaps. Points (1), (2), and (4) above were not addressed. Generally limitations are not discussed. The checklist indicates assumptions were addressed in section 3, but there is no explicit text to this point.


# Clarity

The text can be difficult to read at points, but is generally understandable.

The organization is adequate and the reader is informed of relevant methods and prior work related to the techniques (like double clipping).

Some clarifications and additions that would improve the text:
- What were the instructions given to participants?
- What is the value of the in-game rewards given to participants?
- How was consent obtained from participants?
- (describing supervised learning) "Finally, the macro-goal is predicted via a convergence model of 1 million steps." Not sure what this means. Does this indicate the model trained for 1 million steps and obtained convergence at that point?
- Table 3: How were the human values for different goals determined? For the RL agent we know the conditioned goal but it was not clear how the human goal was decided.
- Consider briefly explaining the Davies-Bouldin index for readers unfamiliar with clustering.


# Significance

The results are modestly important. Researchers looking to condition agent behavior on human examples of strategic choices could adapt the work to other domains. This will be valuable to those training RL agents starting from imitation learning in domains where strategic or other long-term choices are relevant.

The paper advances the state of the art in getting diversity (without compromising quality) for MOBA games in specific and likely for RL game AI agents in general. The comparison to GAIL shows this is superior to at least some alternatives for the same problem.

**Needs Ethics Review:**

Yes

**Time Spent Reviewing:**

2

---

> ### Author Response · Authors · 2021-08-10
> **Response to Reviewer y4xZ**
>
> We appreciate your valuable advice. We will explain your questions below.
>
> Q1: Question about Fig. 7.
>
> A1: MGG models the prior knowledge of the player's strategy, predicts the corresponding macro-goal as the condition, and combines the intrinsic reward to improve the efficiency of the macro strategy exploration. However, we still retain the external reward used in RL-baseline to guide the agent to explore the optimal way to achieve the strategy. Therefore, the macro-state of MGG is closer to humans than that of RL-baseline, but it is still different from humans.
>
> Q2: Question about Fig. 8.
>
> A2: In the case study, all human evaluations are obtained by counting 1,000 human games. In these games, the opponents of the special lineup system are high-rank human players. However, the opponents of RL-baseline, GAIL, MGG are RL-baseline using the regular lineups. Due to the gap in the strength of opponents, there will be differences in indicators between humans and MGG, but MGG is still closer to human than other methods.
>
> Q3: About human-machine test.
>
> A3: Since the players participating in the test include all ranks, the average ability (e.g. micro-operation) is much lower than the RL-baseline. Even esports players of similar strength have been defeated by RL-baseline (Keywords: The Tencent Wukong AI system, The World Champion Cup of 'Honour Of Kings', Kuala Lumpur, August 7, 2019). Almost all testers can only defeat AI through some special strategies, and the values of these strategies are very different from the general reward, so it is difficult to be explored by RL-baseline in self-play training. MGG has learned a lot of human macro strategies and is able to deal with some special strategies. Experiments have also proved that it is slightly better than RL-baseline when facing various human player strategies.
>
> In addition, the human-machine test requires a lot of matches and preparation costs (Line 242), so it is difficult to conduct multiple tests in a short time. However, Fig. 6b shows the cumulative winning percentage. From 12h to 26h, we have counted 15 times for the winning percentage. There is always a gap between the winning percentages of MGG and RL-baseline, and the gap is gradually increasing. It can be ruled out that the difference is caused by statistical errors.
>
> Q4: Clarifications on Meta-Controller convergence.
>
> A4: Yes, we mean that the model has converged at 100w steps, and then we use the model for macro-goal prediction.
>
> Q5: Question about human values in Table 3.
>
> A5: For Resource-Grab System, assassins are generally very strong in the early stage and weaker in the late stage. Therefore, human players usually grab the enemy's monster resources as soon as possible based on the situation of the investigation, thereby opening up the resource gap between the two sides in the early stage. At this time, the goal can be expressed as a combination of moving to the enemy jungle areas, attacking the enemy's monster resources, and raising the level. For Table 3, the value of "Human" is obtained by counting 1000 high-rank human games under the Resource-Grab System.
>
> Q6: Clarifications about the participants.
>
> A6: The instructions given to participants are in-game announcements. The in-game rewards given to participants are the currency in the game and can be exchanged for props. The human-machine test is presented as a challenging activity in the game, which players voluntarily choose to participate in.

---

> > ### Comment · Reviewer_y4xZ · 2021-08-27
> > **response**
> >
> > Thank you for the clarifications and replies.
> >
> > ## From the responses:
> > > However, Fig. 6b shows the cumulative winning percentage. From 12h to 26h, we have counted 15 times for the winning percentage. There is always a gap between the winning percentages of MGG and RL-baseline, and the gap is gradually increasing. It can be ruled out that the difference is caused by statistical errors.
> >
> > 1. To rephrase my question: what statistical test have you used to establish that the two quantities (performance of MGG and RL-baseline) as statistically different? Repeatedly measuring the same quantity at different times is not a measure of statistical error. And the margins are small, so if you intend to report the superior performance I would like to see it expressed in terms of percentage point change and an estimate of error bounds on that difference.
> > 2. The text claims that MGG improves performance. Given the magnitude of this increasing it seems this is at most a slight improvement. I would expect the central claim to instead be that MGG provides more diverse behavior. What is the strong evidence that overall _performance_ improves of the RL-baseline: that is, not evidence that specific corner cases or lineups are addressed, but that in aggregate the agents are winning more / winning "better"?
> >
> > > Since the players participating in the test include all ranks, the average ability (e.g. micro-operation) is much lower than the RL-baseline. Even esports players of similar strength have been defeated by RL-baseline
> >
> > Given this ceiling effect a natural question to ask is what the performance difference between MGG and RL-baseline is when subsetting to highly ranked players. Is there evidence MGG is more effective against strong players?
> >
> >
> > ## From the original review:
> > > The paper is not careful to address weaknesses or remaining gaps. Points (1), (2), and (4) above were not addressed. Generally limitations are not discussed.
> >
> > What are your thoughts on addressing this point? It remains a concern for me.

---

> > > ### Author Response · Authors · 2021-08-30
> > > **Response to Reviewer y4xZ**
> > >
> > > Thank you for your reply again.
> > >
> > > As you suggested, our main goal is exactly making the model to "learn diverse macro strategies" while "maintaining a high level of micro-operations". The secondary goal is to “improve the competitive ability” of the model by increasing the diversity of strategies.
> > >
> > > Therefore, in Sections 4.3 and 4.4, from the overall to the cases, we have proved that MGG has more diverse strategies than the RL-baseline. Regarding the concerns about performance improvement, we will explain in detail.
> > >
> > > Q2-1: About the strong evidence that overall performance improves of the RL-baseline.
> > >
> > > A2-1: We have to clarify that Fig. 6b is just to illustrate that the increase in strategy diversity does not affect the ability when playing against humans. In the human-machine test, the average level of human players is much lower than AI, so human players are not the best object for verifying competitive ability and the advantages brought by the diversity of strategies are not obvious (See [the ELO score including human players](https://sites.google.com/view/mgg-demo/#h.ckykzuvu3b3w), human (1788) < SL (2183) < RL-Baseline (2851) < MGG (3048)).
> > >
> > > Figs. 6(a) and 9 are really strong evidence that the overall performance improves the RL baseline. Fig. 6(a) shows that in the fair match test of randomly selecting heroes, MGG improves ELO by 200 score over the RL-baseline. We need to emphasize:
> > >
> > > > When MGG plays against the RL-baseline, the winning rate is "74%", which results in an increase of 200 ELO score. (See https://sites.google.com/view/mgg-demo/#h.ckykzuvu3b3w)
> > >
> > > This effectively proves that under the premise of similar strength (micro-operation), strategy diversity (with macro goal + intrinsic reward) is conducive to increasing the ceiling of the competitive ability.
> > >
> > > Q2-2: Statistical test and the new presentation of Fig. 6(b).
> > >
> > > A2-2: We re-display the results of Fig. 6(b) in two ways, namely, the match win rate per hour (a total of 26 test groups), and the detailed accumulative win rate. We also perform F-test and t-test on the "Match Win Rate Per Hour". The result shows that at the significance level of 0.05, there is a significant difference between the average win rates, and the average win rate of the RL-baseline is significantly lower than that of MGG. (See https://sites.google.com/view/mgg-demo/#h.1s2tagd7rzm7)
> > >
> > > Q2-3: The evidence that MGG is more effective against strong players.
> > >
> > > A2-3: The appendix ([Sections A.1.2, A.1.3](https://sites.google.com/view/mgg-demo/#h.czpouyumsesw)) shows some tests with high-rank players under some special lineup systems. The metrics against top human players show that MGG has a significant improvement over the RL-baseline.
> > >
> > > Q2-4: About the thoughts on (2).
> > >
> > > A2-4: We explained it in the original A1. The intrinsic reward can guide the model to achieve human macro goals at every C moment. However, in time (t, t+C), the external reward will still be used to guide the model to explore the way to achieve, resulting in different macro-states from humans. This issue can be considered as another problem, that is, personification in MOBA games. In this text, we will not discuss it in depth.
> > >
> > > > A1: MGG models the prior knowledge of the player's strategy, predicts the corresponding macro-goal as the condition, and combines the intrinsic reward to improve the efficiency of the macro strategy exploration. However, we still retain the external reward used in RL-baseline to guide the agent to explore the optimal way to achieve the strategy. Therefore, the macro-state of MGG is closer to humans than that of RL-baseline, but it is still different from humans.
> > >
> > > Q2-5: About the thoughts on (4).
> > >
> > > A2-5: We explained it in the original A2. Due to the high cost of human-machine test for each special lineup system, human indicators are all calculated from human games. The difference of opponents leads to differences in the game state (e.g., kill, dead, level, gold, etc.), and the macro-goal predicted by the Meta-Controller will also be different, so there is a gap in indicators. But this is not a limitation of methodology. For the same opponent, MGG is still superior to the RL-baseline and GAIL.
> > >
> > > > A2: In the case study, all human evaluations are obtained by counting 1,000 human games. In these games, the opponents of the special lineup system are high-rank human players. However, the opponents of RL-baseline, GAIL, MGG are RL-baseline using the regular lineups. Due to the gap in the strength of opponents, there will be differences in indicators between humans and MGG, but MGG is still closer to human than other methods.

---

### Official Review · Reviewer_uPYF · 2021-07-10

**Rating:** 5
**Confidence:** 5

**Summary:**

The authors propose a method for using limited human data to generate macro-goals that serve as an auxiliary imitation loss and reward for deep reinforcement learning agents playing a mobile 5v5 MOBA game (Honor of Kings). The proposed approach provides 'macro' supervision for agent behavior while letting the agent learn the finer details and micro actions to reach goals or maximize game reward.
Experiments compare the proposed approach to a variety of baselines and indicate higher ELO scores and lower exploitability by human players.

**Ethics Review Area:**

["I don’t know"]

**Limitations And Societal Impact:**

# Limitations

The authors do not describe specific limitations of their work. In particular use of human data can be seen as a limitation as compared to other MOBA efforts. Furthermore the introduction of additional rewards and losses may complicate or destabilise training, and this should be addressed or discussed.

# Societal Impact

The authors properly address the societal impact of their work in the checklist and the main text.

**Main Review:**

# Overview

"Learning Diverse Policies in MOBA Games via Macro-Goals" presents a compelling direction for combining offline human feedback with a reinforcement learning policy, and the evaluation criteria investigating exploitability is novel and supports the paper's stated goals. Certain areas of the paper require work to help the reader reproduce the results or understand the significance of the outcomes.

# Originality

The paper's combination of human data using imitation learning and reinforcement learning has been previously explored in the context of 'human preferences' and summarization work. The approach is novel in the context of MOBA games and demonstrates that in long-action sequences this form of supervision is applicable and can lead to demonstrably more diverse policies.

The choice of neural network, features, domain, and other aspects are not the focus of the paper, and are also not novel.

The paper measures the exploitability of the agents using human evaluations (Figure 6) and this evaluation format has not been used previously in the context of learning game AI policies. This test, while difficult to reproduce, serves as an invaluable showcase for the method's impact on realistic play.

# Clarity

The paper's stated goals and application domain is clearly stated. Similarly, the purpose of the experiments is explained and motivated. Certain results and their applicability hinder the reading and contextualization of the work. As an example, 4.4.3 discusses capability analysis while using ELO scores. The compared approaches are stated to be 'converged', however this term is loaded and hard to transfer across architectures and setups. Similarly the results in Figure 6 and 9 do not provide statistical significance of the results, or potentially a comparison across different points during training (as a way of further confirming that the systems have indeed converged). As such, the conclusions from many of these experiments are hard to measure. A key question that should be answered is: "does using human games and an additional goal reward in this way provide better results provided with the same resources/time/energy as our reinforcement learning baseline?". Ideally a good control would allow the reinforcement learning policy to train supervised from human trajectories and then train with self-play, and account for the additional parameters and training time required by the imitation training.

Section 4.4.1 Why is achieving "resource intent strategies" important? Does the trained policy *need* to be good at this to improve at the game, or is the purpose to be as similar to human play?

Section 4.4.2 discusses human performance for monster resources: what is this human evaluation? how are they chosen? How many attempts are done by humans? Are these humans professional or amateur players?

# Significance
The proposed approach is tested on a challenging benchmark and uses significant resources to train the agents and evaluate them.
The significance of this work is limited by two key factors: 1) research in this area is important, but the proposed technique is hard to reproduce owing to the high computational cost, lack of a public interface or system to test and train Honor of Kings agents. 2) the results are provided without statistical significance numbers or selection criteria for human players. The ELO numbers and reward scores should ideally be amended in the final version of the paper with some measure to understand the size of the gains over GAIL or RL-Baseline. The human player performance is hard to asses, and perhaps an ELO score for human players, or a description of the selection process would help. (specifically the exploitability plot in Figure 6 strongly depends on the kind of players that tested either agent, and whether they were allowed to play against either system, or only one kind repeatedly, etc.)

# Areas for improvement

## Statistical significance:
Figure 6 and Figure 9 provide the most impressive results of the paper, but lack any statistical significance metrics which would help understand the gains between RL-baseline, GAIL, and MGG.

## Wording:
L249 "this test effectively proves" I would reword this as "this test suggests" or "provides evidence for" as conclusive demonstrations with human testing and a limited number of samples are unlikely, but do provide a great benchmark.
L259, L265 "marco states" -> "macro states"

## Citation style:
OpenAI Five citation should preferably use https: //openai.com/bibtex/openai2019dota.bib (e.g. OpenAI et al.)

## Related Work:
L78 "All the mentioned works except the SL model adopt a single strategy regardless of the
79 heroes and lineups, which cannot take full advantage of heroes and lineups’ characteristics" -> Slight correction: OpenAI Five has a latent vector encoding lane preferences + a representation of the heroes that affect the employed strategy, so in practice the strategy will change dramatically between games when those aspects are randomized.

**Time Spent Reviewing:**

12

---

> ### Author Response · Authors · 2021-08-10
> **Response to Reviewer uPYF**
>
> Thank you very much for your valuable advice and we will definitely improve the presentation and result description soon. For your questions, modified expressions, and experimental supplements, we will show below.
>
> Q1: About model capability convergence in section 4.4.3.
>
> A1: To make a fair comparison, we use exactly the same framework, settings, computational resources to train RL-baseline, GAIL, and MGG. Other than that, the only difference is that MGG = RL-baseline + macro goal (feature) + intrinsic reward, GAIL = RL-baseline + discriminator + adversarial reward. In order to judge the model is ‘converged’, we use the models at different time points to play against each other every day, and then calculate the ELO until it does not change. Therefore, it can be considered that under the same resources, the 'converged' model has reached the upper limit of its ability. Due to length limitations, we did not show the ELO results at different time points during training.
>
> Q2: Questions about the results of ELO and the gains compared to other methods.
>
> A2: As shown in Line 238, we use all methods to play pairwise matches, 400 games each pair, and then calculate the ELO scores. All lineups in each match are randomly selected from the 102 hero pool, that is, any lineup system. For the rigor of the experiment, we used different random seeds to select heroes and supplemented 20 experiments. The error range of the final ELO is ± 30 points. Therefore, the ELO score can indicate the gain of MGG over GAIL and RL-baseline under different lineup systems.
>
> In addition, the appendix (Sections A.1.2, A.1.3) also shows some tests with high-rank players. It can be seen that under some special lineup systems, MGG has a significant improvement over RL-baseline.
>
> Q3: Question about computational resources and time.
>
> A3: Our main goal is for the model to learn diverse macro strategies while maintaining a high level of micro-operations (Line 65-67). And the secondary goal is to improve the competitive ability of the model by increasing the diversity of strategies (Line68-70). Therefore, we are more focused on whether using the same computational resources can improve the model's strategy diversity and competitive ability, rather than time. Under the same computational resources, the increase in training time cannot continue to improve the ability of RL-baseline, so the ELO comparison of the converged models provides evidence for the effectiveness of MGG.
>
> Q4: The importance of "resource intent strategies" in Section 4.4.1.
>
> A4：In the Marksman-Core System, there is a very powerful marksman. In order to maximize the ability of the lineup, human players will allocate more resources to the marksman to open up the advantage, but RL-baseline will not. Therefore, the policy can only improve in the game if it remains similar to humans in terms of resource intent.
>
> Q5: Question about the human evaluation for monster resources in Section 4.4.2.
>
> A5: We have counted a total of 1,000 human games using the Resource-Grab System. Both sides of the game are high-rank players in Honor of Kings.
>
> Q6: Question about competitive performance evaluation of human players.
>
> A6: The human-machine test is open to all players in Honor of Kings, from amateur to professional players. And MGG and RL-baseline are randomly selected as opponents (Line 243-245). Both human players and AI can choose any hero from the 102 hero pool, and humans can use any lineup system to fight against the AI. In addition, we also conducted 50,000 human-machine tests on the SL model, and the winning rate against humans was 93.10%. Therefore, we can also calculate the ELO scores for human players, and the final score is no-op (0) < Built-in bot (959) < human (1788) < SL (2183) < RL-Baseline (2851) < MGG (3048).
>
> Q7: About Wording, Citation style, Related Work, Limitations.
>
> A7: We will modify it in subsequent versions.

---

### Official Review · Reviewer_S81z · 2021-07-14

**Rating:** 5
**Confidence:** 4

**Summary:**

The paper proposes a novel method for learning policies to play MOBA (Multi-player Online Battle Arena) games which makes use of expert supervision to extract and learn the prediction of high-level or macro goals (meta-controller) which is subsequently used to train a policy using reinforcement learning to execute each goal with a low-level micro-controller. The proposed method has an emphasis on diversity in strategies which is built-in by sampling from the meta-controller and is also evaluated with quantitative and qualitative experiments. This diversity in strategies is stated to be the reason for improvement over human performance when compared to state of the art baselines. Several qualitative demonstrations of the proposed method (Section 4.4) show how non-conventional strategies are adopted more easily than baselines. A human vs machine study is also presented for measuring the win-rate of the proposed method and baselines and how it changes with more test time i.e. how the win-rate changes when humans are able to learn and improve their own performance against the AI matchups.

**Limitations And Societal Impact:**

Yes, the authors have addressed the limitations of the paper adequately. They should also cite Ye et al. [2020a] for the detailed broader impacts that they have mentioned for MOBA game based AI.

**Main Review:**

Overall, the paper presents a novel method for pushing the state of the art of AI performance in MOBA games. The large-scale human study to evaluate performance against baselines and the performance exceeding the state-of-the-art baselines clearly demonstrate the effectiveness of the proposed MGG method. Specific analyses also support the diversity claims made about the learned strategies. However, the paper in its current state leaves a lot to be desired in terms of clarity, broader motivation and accessibility. Despite the thorough empirical analysis made in this paper, it remains inaccessible to the reader due to these writing issues.

Below are detailed comments on the strengths and weaknesses that informed my evaluation of the paper.


## Strengths

1. The presented MGG method’s main selling point is the diversity in strategies that are aimed to improve performance against human matchups, since humans are capable of overcoming unimodal/repetitive strategies by AI agents given enough time and practice. The claim for diversity in the proposed method is supported by a quantitative analysis with measurements of goal-entropy and Davies-Bouldin Index of the goal embeddings, and a qualitative analysis of specific cases of non-conventional strategies in the chosen game (Honor of Kings) which typically do not align with the default reward models proposed for this game.

2. The large scale human-study with human vs AI matches clearly demonstrates that the proposed method retains performance (w.r.t the baseline) with increasing human test time, which is in line with the hypothesis that diverse strategies are better than unimodal strategies against human matchups.

3. The choice of using expert demonstrations for learning only the high-level macro-goal prediction network is intuitively appealing -- it aims to mimic broader human intentions without worrying about matching the exact execution of each intention.

4. For readers that are familiar with MOBA games, the two case studies in Sections 4.4.1 and 4.4.2 (and the videos linked in the paper) are impressive demonstrations of what the proposed method is capable of in practice. Such strategies seem to be very difficult to learn for AI agents but are easy or obvious for human players.



## Weaknesses


#### Lack of clarity in method description.

1. The second paragraph in Section 3 is confusing due to lack of proper explanation and use of arbitrary new notation. The difference between macro states and macro goals is not clear. The role of f i.e. the macro state extraction function is not made clear until much later in the paper. The “general reward function” is never explained -- what does this mean and what does it consist of? What does the “common strategy” mean? Why is notation introduced for human macro states when this notation is never used again in the entire paper (i.e. served no purpose later on). It could have been explained without any notation. In my opinion, the goal space should be completely defined at this stage of the paper, along with the choice of goal dimension and a description of what the dimensions of the goal specify. Also, notation for intrinsic reward R(f(S), G, A) is introduced without being accompanied by a definition (the definition is much later in the paper).

2. Auxiliary tasks are first mentioned in L135 but what auxiliary tasks are used is not mentioned until L173. Since only a single auxiliary task of auto-encoding is used, this explanation does not deserve to be pushed to such a later stage from it’s first mention.

3. L161 states that C is the number of frames between state and its goal, but it is not made clear how C is specified. Later in the paper, C is then stated to be a hyperparameter which has a fixed value. This description about C being a hyperparam of choice and its value should all be in the same first mention of C in the paper.

4. Equation 2 (and L192 below it) assume that the goal space admits a metric in the form of an L1 norm, but the goal space is not defined at this point in the paper. There is an implicit assumption that the goal space is a multi-dimensional euclidean space (R^d).


#### Accessibility.

1. The paper does not introduce key MOBA-specific terminology before using it at several points in the paper. A basic explanation of the mechanics, heros and progression milestones of the game should be explained before terms like “destroying the crystal”, “over 102 heroes…” or “ban/pick capabilities” are used. However, given that this paper specifically focuses solely on MOBA games, this is a minor issue.
The paper mentions “league based” methods in the intro (L43 and L66) but does not explain what this category of prior work means until L81, where a very vague explanation is presented -- “league-based methods train multiple agents simultaneously to explore strategy space and require massive computation”.

2. The “general reward function” is mentioned several times in the paper but only introduced on L274 after citing Ye et al. [2020a] without any description. This seems to be an important fact to include in the paper well in advance.
The fact that the proposed method is an instance of an imitation learning method where expert data is used is not made clear during comparisons with baselines. It is not mentioned that the “RL-baseline” method does not use any human supervision, which is an important point to note in the comparison.


#### Empirical issues.

1. For the main result of AI vs human win rate obtained from the large scale human study, the Y-axis range in Figure 6 (b) seems to be extremely narrow (98.72% - 98.86%). Due to the absence of error bars, it is very difficult to judge the statistical significance of the presented results. It is also very surprising that the RL-baseline, despite lacking diversity, is able to retain such a high win rate after 26 hours of accumulated human test time. This seems to strongly oppose the premise of the paper that humans can easily overcome AI strategies if they lack diversity.

2. Lack of error bars: Figures 6, 9 and Tables 1,2,3 lack error bars, despite the authors having answered “Yes” to the checklist question about whether they have included error bars.


## Other minor errors

1. L259 -- A brief explanation of the Davies-Bouldin Index and its relevance should be presented instead of just citing the original paper.
2. Last line of Conclusions seems very vague, what does “non-transitivity in strategies” mean? The directions for future work can be made more clear.
3. Table 2 and Section 4.3.2 mention measuring “state entropy”. Shouldn’t this be called “macro state entropy” instead?


**Time Spent Reviewing:**

5

---

> ### Author Response · Authors · 2021-08-10
> **Response to Reviewer S81z**
>
> Thank you for your valuable suggestions. We feel sorry for the lack of a clear description of the definition and explanation of the symbol, and we will definitely be able to modify it based on comments in a short time. Below are some demos we modified, hoping to share our work with the MOBA game AI community.
>
> Q1：Modified demo of Section 3.
>
> A1: According to your suggestion, we add a detailed description of the symbol. Then, explain the knowledge related to RL-baseline earlier.
>
> > In particular, we introduce $G$ which represents the macro strategic states space in the game, which is a multi-dimensional Euclidean space. And we also define a macro state extraction function $f:S \rightarrow G$ for mapping the current state $s$ to the macro state $c=f(s)$. The macro state $c \in G$ mainly includes the region state (e.g. the area on the map) and resource state (e.g. unit resource, gold, level) related to strategy. The macro goal $g \in G$ represents the macro state that the current strategy should achieve in the future.
>
> > Ye et al. [2020a] design the general reward function for MOBA games, including farming, KDA, damage, pushing, and win/lose related rewards. As a result of the single value, the policy only learns the common strategy from self-play training, that is, farming first, then fighting and pushing, no matter what strategy the opponents use, so its macro-state set {$c|c \in G_{RL}$} cannot fully cover $G$, i.e. $G_{RL} \subset G$. However, human strategies are diverse, so the human macro-state set $G_{human}$ is considered to be close to the complete $G$, i.e. $G_{human}=G_1 \cup \dots G_i \dots \cup G_N \approx G$, where $N$ is the number of human strategies. We define that strategy $i$ can be expressed as a set of macro goal {$c|c \in G_i$}, where $G_i \subset G$. Therefore, we model the policy as $\pi(a|s, g)$, and our objective is to use $g \in G_{human} \approx G$ as the macro goal of the policy and use intrinsic reward $R(f(S), G, A)$ to guide the macro state of the policy to be close to the complete $G$, that is, to maximize the marco state entropy $-\sum_{c \in G}\ P(c)log P(c)$. We define the intrinsic rewards $R(f(S), G, A)$ as the delta of the distance between the current macro state $c$ and the macro goal $g$. If the macro state $c_{t+1}$ is more close to macro goal $g_{t}$  than $c_t$, the policy will receive a positive reward.
>
> > During the training, we introduce the auxiliary auto-encoding tasks $T_{aux}$ of lineups to improve prediction performance.
>
> > Therefore, we can sample the frame with goals from the interval $[T_{s}+C-\epsilon, T_{s}+C+\epsilon]$, where $C$ is a hyperparameter representing the delta of frame numbers between the state and its goals, generally 30s.
>
>
> Q2: About the human-machine test.
>
> A2: Since the players participating in the test include all ranks, the average ability (e.g. micro-operation) is much lower than the RL-baseline. Even esports players of similar strength have been defeated by RL-baseline (Keywords: The Tencent Wukong AI system, The World Champion Cup of 'Honour Of Kings', Kuala Lumpur, August 7, 2019). Almost all testers can only defeat AI through some special strategies, and the values of these strategies are very different from the general reward, so it is difficult to be explored by RL-baseline in self-play training. MGG has learned a lot of human macro strategies and is able to deal with some special strategies. Experiments have also proved that it is slightly better than RL-baseline when facing various human player strategies.
>
> In addition, the human-machine test requires a lot of matches and preparation costs (Line 242), so it is difficult to conduct multiple tests in a short time. However, Fig. 6b shows the cumulative winning percentage. From 12h to 26h, we have counted 15 times for the winning percentage. There is always a gap between the winning percentages of MGG and RL-baseline, and the gap is gradually increasing. It can be ruled out that the difference is caused by statistical errors.
>
> It can also be seen from the ELO test in Fig. 6a that in the match test of randomly selecting heroes, MGG improves ELO by 200 points over RL-baseline. This effectively proves that under the premise of similar strength (e.g. micro-operation), strategy diversity (with macro goal + intrinsic reward) is helpful to increase the ceiling of competitive ability.
>
> Q3: About error bars.
>
> We will add error bars in subsequent versions. And for the rigor of the experiment, we used different random seeds to select heroes and supplemented 20 experiments. The error range of the final ELO is ± 30 points.
>
> Q4: Questions about the key MOBA-specific terminology.
>
> A4: Due to space limitations, we only briefly explain the state, action, win/lose (Line 111-114), and map resources (Fig. 2) of the MOBA game in the paper. We will make adjustments in subsequent versions. For example,
> > In a MOBA game competition, both players enter the ban/pick stage first, and select heroes from the hero pool to form a lineup.  Then, the players use the selected lineup to continuously obtain resources (farming) in the game, increase gold coins and levels, kill the enemy (fighting) and destroy the turret (pushing), and finally destroy the crystal to win.
>
> Q5: The modified demo of "league-based" methods in Line 43.
>
> A5:
> > AlphaStar Vinyals et al. [2019] uses a framework based on the league to improve the diversity of policies in StarCraft II. They construct the league which is initialized by supervised learning, including main agents, main exploiters, and league exploiters. The main agents train against all of these past players, as well as themselves. The league exploiters train against all past players. The main exploiters train against the main agents. The role of the exploiters is to highlight flaws in the main agents forcing them to discover new strategies. These methods increase a huge computational and time cost to training.
>
> Q6: About the minor errors.
>
> A6: We will definitely make improvements in subsequent releases.

---

### Official Review · Reviewer_e26o · 2021-07-16

**Rating:** 6
**Confidence:** 4

**Summary:**

This paper proposes improvements to an existing deep RL agent for the game Honor of Kings, by introducing a meta-controller trained by supervised learning to predict macro goals high-level players would like to pursue next.

**Ethical Concerns:**

This paper presents results from both:

- training the meta-controller on human gameplay data;
- evaluating the agent in games with human players

But has limited documentation of how (or if) players provided informed consent to participate. Lines 242-250 imply players in the evaluation phase were made aware they were playing an AI and were rewarded in-game for doing so (noted on the checklist) but I see no evidence that players who's data was used to train the meta-controller gave permission for this use.

**Ethics Review Area:**

["Responsible Research Practice (e.g., IRB, documentation, research ethics)"]

**Limitations And Societal Impact:**

There is limited discussions of limitations in the paper, but plenty that could be discussed to improve the paper:

(1) Elo scores are used throughout the paper to evaluate the agent, without acknowledgement of known issues with this form of evaluation when games (including MOBAs) have non-transitive interaction between policies. See for example Balduzzi et al. Re-evaluating Evaluation [NeurIPS 2018]

(2) Due to the scale of the environment, there are limited ablations and repeats. This is an existence proof of the ability for agents to play at this level, and not a rigorous study or the robustness of this approach. I think this is an acceptable form of publication, but this detail should be acknowledged.

**Main Review:**

The originality of the method is limited - hierarchical control systems are well established and games of similar complexity and with similar mechanics have been learnt previously - but it remains interesting to the sub-community focused on these types of applications as additional evidence of approaches that can be successful utilized in modern games.

I have some concerns about the correctness of statements in the paper that could significantly improve my rating of this paper if adequately addressed in the author's reply:

(1) Figure 6b shows win rate against human without variance. Given the very small differences in percentages on the y-axis, there may not be any significant improvement over the previous RL baseline. Can you provide stronger evidence that the proposed method significantly improves performance in comparison to the chosen baseline?

(2) The contributions on page 2 state the proposed method uses the same computational resources as the baseline, but the proposed method introduces an additional pre-training stage for the macro classifier and augments the policy state space. Both of these modifications will require additional compute - the macro classifier in its own training and the augmented policy space I assume will increase training time for the RL agent. Should this statement instead be that once the agent is trained they use the same resources at the evaluation stage? If so, the claim that this is "unlike league-based methods" does not hold.

(3) On line 40 it is claimed that "the agent should exploit [the opponent] instead of playing with a single policy" but these two statements are not mutually exclusive. A single policy can learn to exploit any opponent, either by playing a Nash equilibrium strategy or using opponent modelling to recognize online the type of opponent it is playing and condition its policy on this additional observation.

(4) Line 85 claims "there is no obvious signal ... to represent macro strategies in MOBA games" as a way to differentiate from prior work but the proposed method defines macro goals (section 3.1.1), uses them throughout training and claims they are applicable to other MOBA games (line 143). I think there are other ways to differentiate the proposed method from the past work cited, but this current statement is contradicted by the work done and so currently weakens the evidence of originality presented in the paper.

Other issues that are lowering my current rating of the quality and clarity of the paper:

(5) Line 232-233 gives statistics about the agents response time, but provides no grounding for how this compares to human players.

(6) In Figure 9 do the built-in bots have an ELO rating of 0? Or is this the no-op agent as in Figure 6?

Minor comments that could be easily addressed if accepted (no need to respond):
- Due to the complexity of the approach and scale of the study, some details are missing that would be interesting insights to the overall system. Most notably, no results are provided for the meta-controller pre-training and the focal loss is not defined (presumably because it is defined in Lin et al. 2017) - adding these to the appendix would make this a more self contained reference.
- Line 190: "even the lineups" -> "even WHEN the lineups"
- Line 263: "the marco state" -> macro

**Needs Ethics Review:**

Yes

**Time Spent Reviewing:**

4

---

> ### Author Response · Authors · 2021-08-10
> **Response to Reviewer e26o**
>
> Thanks for your constructive comments. For your concerns, we will explain in detail below.
>
> Our main innovation is to propose a general hierarchical control framework for MOBA games to learn diverse policies, which is an unsolved MOBA-game-playing problem. The main goal is for the model to learn diverse macro strategies while maintaining a high level of micro-operations (Line 65-67). The secondary goal is to improve the competitive ability of the model by increasing the diversity of strategies so that the model is not easily countered by a single strategy, and can play the most suitable strategy for the selected lineup.
>
> Q1: Evidence of significant improvement in performance.
>
> A1: Firstly, for your doubts about Fig. 6b, our explanation is as follows.
>
> - Human-machine test requires a lot of matches and preparation costs (Line 242), so it is difficult to conduct multiple tests in a short time. However, Fig. 6b shows the cumulative winning percentage. From 12h to 26h, we have counted 15 times for the winning percentage. There is always a gap between the winning percentages of MGG and RL-baseline, and the gap is gradually increasing. It can be ruled out that the difference is caused by statistical errors.
> - Since the players participating in the test include all ranks, the average ability (e.g. micro-operation) is much lower than the RL-baseline. Even esports players of similar strength have been defeated by RL-baseline (Keywords: The Tencent Wukong AI system, The World Champion Cup of 'Honour Of Kings', Kuala Lumpur, August 7, 2019). Almost all testers can only defeat AI through some special strategies, and the values of these strategies are very different from the general reward, so it is difficult to be explored by RL-baseline in self-play training. MGG has learned a lot of human macro strategies and is able to deal with some special strategies. Experiments have also proved that it is slightly better than RL-baseline when facing various human player strategies.
>
> Then, other strong evidence in the paper also shows that we have reached the goal:
>
> - From the perspective of strategy diversity (main goal), MGG is significantly better than RL-baseline (Sections 4.3 and 4.4)
> - From the perspective of competitive ability:
>
>     (1) A large amount of human-machine test data shows that MGG is slightly improved compared to the RL-baseline (Fig. 6b);
>
>     (2) The appendix (Sections A.1.2, A.1.3) also shows some tests with high-rank players. It can be seen that under some special lineup system, MGG has a significant improvement over the RL-baseline;
>
>     (3) It can also be seen from the ELO test in Fig. 6a that in the match test of randomly selecting heroes, MGG improves ELO by 200 points over the RL-baseline. This effectively proves that under the premise of similar strength (e.g. micro-operation), strategy diversity (with macro goal + intrinsic reward) is helpful to increase the ceiling of the competitive ability. For the rigor of the experiment, we used different random seeds to select heroes and supplemented 20 experiments. The error range of the final ELO is ± 30 points.
>
> Q2: Question about computational resources.
>
> A2: First, the training of MGG is divided into two steps. As mentioned in Line 225, for the pre-training of Meta-Controller, we only need 8 NVIDIA P40 GPUs, and the training can converge for 26 hours. Compared to the training of RL-baseline (Line 230-231, a physical computing cluster with 230 60,000 CPUs and 830 NVIDIA V100 GPUs, 420 hours), time and computational resources are very low.
> Then, when training the policy network, the Meta-Controller is not trained, and the parameter amount of 1.7 million is insignificant (1%) for 174 million parameter amount of the original policy network (RL-baseline). For the 64 dimensional macro goals, we only increase the parameter amount by 50W (0.2%). In addition, compared with the RL-baseline (45ms), the inference time of MGG for a round (47ms) will only increase by only 2ms (4.4%).
>
> Finally, the rest of the training configuration is exactly the same as the RL-baseline, and it is learned by randomly selecting heroes and self-play training. Therefore, compared with league-based methods, the increased resources are minimal. Moreover, we cannot learn SL models separately for all lineups of MOBA games to build opponent pools, because the cost is huge.
>
> Q3: Description about "the agent should exploit ... instead of playing with a single policy" in Line 40.
>
> A3: I'm sorry for misusing the word "exploit". What we originally meant is that the model should "find a way" to maximize the strength of the chosen lineup when playing with different lineups, rather than the meaning of "exploit the opponent".
>
> Furthermore, because RL-baseline uses the general reward function (Ye et al. [2020a]) and is trained through self-play, it is difficult for the model itself or the opponent to learn diversity strategies under common values. (Section 4.3, 4.4).
>
> Q4: Description about “there is no obvious signal... to represent macro strategies in MOBA games " in Line 85.
>
> A4: This is also a lapse in the description. We don't mean that "no obvious signal in MOBA", rather, it is difficult to use artificial building orders to represent strategies like Starcraft II, and there are dependencies between buildings. The strategy of the MOBA game can be composed of a large number of changeable macro-goals. Therefore, our originality is to use human data to learn the distribution of macro-goals under different states to represent strategies, which is the difference from the previous work.
>
> Q5: Question about reaction speed of human players.
>
> A5: The averaged APMs of our AI and top players are comparable (80.5 and 80.3, respectively). The proportions of high APMs (APM ≥ 300 for Honor of Kings ) during games are 4% for top players and 5% for our AI, respectively.
>
> Q6: Question about built-in bots ELO score in Fig. 9.
>
> A6: We did not add the no-op agent in Fig. 6 to Fig. 9. The built-in bots fail against all other models, so the ELO is 0. We will show more aesthetically in subsequent versions.
>
> Q7: About Ethical Concerns.
>
> A7: For the training data of Meta-Controller, we get it through game-licensed race data.
>
> Q8: About Limitations and Societal Impact.
>
> A8: We will definitely make improvements in subsequent releases.

---

> > ### Comment · Reviewer_e26o · 2021-08-28
> > **RE: Response**
> >
> > Thank you for the detailed response. I accept your replies to Q2-6 and Q8 but have outstanding concerns about Q1 and Q7 that I will discuss below. I hope that you will use the feedback here from all reviewers and your responses to improve the clarity of the paper if accepted.
> >
> > **Q1:** There remains uncertainty amongst the reviewers whether there is sufficient evidence that performance has significantly improved. An alternative interpretation of these results could be that the methods proposed significantly improve diversity of behaviour but make no significant difference to performance. This interpretation counters the assumed hypothesis of the paper that improving diversity will cause an improvement in performance. This could be a very interesting result of wide interest to the community as it also differs from established expectations for these types of games. However, the current uncertainty in which interpretation is correct prevent me from being able to support acceptance. Can you make a stronger argument for either case to remove this uncertainty?
> >
> > **Q7:** I am concerned by your response to ethics reviewer GhhQ that no IRB process was followed because the game had already been released. Can you confirm this was your intended response? Prior release of a commercial product does not make researchers exempt from following ethical procedures in their experiments on the user base of the product.

---

> > > ### Author Response · Authors · 2021-08-30
> > > **Response to Reviewer e26o**
> > >
> > > Thank you very much for your reply. Your suggestion is very meaningful to us.
> > >
> > > Q2-1：Strong evidence of significant improvement in performance.
> > >
> > > A2-1: I'm very sorry, for this uncertainty, which may be caused by the way of presentation. In order to more reasonably prove that increasing the diversity of strategies can improve the performance, we will show the conclusion of Fig. 6 in [two new ways](https://sites.google.com/view/mgg-demo/#h.ckykzuvu3b3w).
> > >
> > > First, we will show the raw win rate comparison used to calculate ELO in Fig. 6, rather than the ELO comparison. Obviously, MGG is not only a “200 ELO score” higher than the RL-baseline, but the win rate against the RL-baseline is “74%”. This intuitively shows that strategy diversity (with macro goal + intrinsic reward) is conducive to increasing the ceiling of the performance.
> > >
> > > |       　      | No-op | Built-in Bots | Human player |      SL      | RL-baseline |     MGG     |
> > > |:-------------:|:-----:|:-------------:|:------------:|:------------:|:-----------:|:-----------:|
> > > |     No-op     |   -   |       0       |       0      |       0      |      0      |      0      |
> > > | Built-in Bots |  100  |       -       |       0      |       0      |      0      |      0      |
> > > |  Human player |  100  |      100      |       -      |      6.9     |     1.23    |     1.19    |
> > > |       SL      |  100  |      100      |     93.1     |       -      | 0.01 ± 0.01 | 0.01 ± 0.01 |
> > > |  RL-baseline  |  100  |      100      |     98.77    | 99.99 ± 0.01 |      -      |  26.09 ± 3  |
> > > |      MGG      |  100  |      100      |     98.81    | 99.99 ± 0.01 |  73.91 ± 3  |      -      |
> > >
> > > Then, we also calculate the ELO scores for human players in the human-machine test, and the final score is no-op (0) < bot (959) < human (1788) < SL (2183) < RL-Baseline (2851) < MGG (3048). The average level of human players is much lower than AI, so human players are not the best object for verifying competitive ability. The ELO score after including human players can clearly reflect the performance improvement.
> > >
> > > Similarly, the ELO score of Fig. 9 also shows that MGG has a significant improvement over the RL-baseline and GAIL. The metrics against top human players in the appendix ([Sections A.1.2, A.1.3](https://sites.google.com/view/mgg-demo/#h.czpouyumsesw)) also verified this conclusion.
> > >
> > > Q2-7: About IRB process.
> > >
> > > What we really meant is that "Honor of Kings" is a released and recognized "testbed" for MOBA-game-playing problems.  Ye et al.[2020b,c,a] and Wu [2019] both used "Honor of Kings" as the "testbed" for research, including tests with top players. We consulted them and implemented similar measures and agreements to protect the welfare, rights, and privacy of human subjects.
> > >
> > > We have performed a process similar to IRB before the test conducted in this paper. Our research has been approved by our institution and all participants.
> > >
> > > - First, we analyze the risks of these experiments to the participants. The risks mainly include the leakage of identity information and the time cost.
> > >
> > > - Then, a series of measures are implemented to prevent these risks. We make a risk statement for participants and sign an identity information confidentiality agreement. We only use information related to the game state in our research, without any identity information. In addition, special equipment and accounts are provided to the participants to prevent leakage of equipment and account information during the test. The identity information of all participants is not disclosed to the public.
> > >
> > > - Finally, the purpose of all tests is only for academic research. Our research will make contributions to the research community, the game industry, and the esports community. (See A2 of Reviewer GhhQ)

---

> > > > ### Comment · Reviewer_e26o · 2021-09-02
> > > > **Score Updated**
> > > >
> > > > Thank you for the detailed response (both now and throughout the discussion period) this significantly improves my view of the paper. I have updated my score to reflect this and encourage you to revise the paper to incorporate this clearer demonstration of the improved performance.

---

### Review · Ethics_Reviewer_GhhQ · 2021-08-11

**Recommendation:**

I recommend the authors do the following:

1) Add more detail about the human-machine test in the appendix:

a) provide the test instructions in the game

b) include a link to the IRB approvals and discuss potential concerns that IRB raised

c) explain how much in-game rewards were given out and show the distribution of these rewards

2) Discuss the potential societal consequences of this line of research. For instance, in the US, DARPA is looking into improving machine performance in tactical war games. I think ML research using MOBA games could easily be applied in this military context.
https://www.militaryaerospace.com/computers/article/14177043/artificial-intelligence-ai-war-games-simulation


**Ethical Issues:**

Yes

**Ethics Review:**

The two concerns I have are:
1) the authors have not presented IRB documentation when they have conducted research with human players
2) the authors have not discussed the potential use of their line of research for military wargaming exercises

---

> ### Author Response · Authors · 2021-08-14
> **Response to Ethics Reviewer GhhQ**
>
> Thank you very much for your suggestions. We will explain your concerns and add relevant information in the appendix.
>
> Q1: Detail about the human-machine test.
>
> In order to ensure the double-blind review, we will not show detailed information here.
>
> - As a game activity, we will clearly indicate that this is an activity to challenge AI at the “entry” of the activity. In addition, near the "Start Challenge" button, we will also describe the number of in-game currency rewards that can be obtained by participating in the activity. These in-game currencies are usually used to redeem items in the game. The activity has no effect on the normal game process. This means that players can voluntarily choose whether to participate in the activity. This activity is extra entertaining for human players.
>
> - "Honor of Kings" is a released and recognized "testbed" for MOBA-game-playing problems.  Ye et al.[2020b,c,a] and Wu [2019] both used "Honor of Kings" as the "testbed" for research, including tests with top players. We consulted them and implemented similar measures and agreements to protect the welfare, rights, and privacy of human subjects.
>
> - For human e-sports players, we have performed a process similar to IRB before the test conducted in this paper. Our research has been approved by our institution and all participants.
>
>     - First, we analyze the risks of these experiments to the participants. The risks mainly include the leakage of identity information and the time cost.
>     - Then, a series of measures are implemented to prevent these risks. We make a risk statement for participants and sign an identity information confidentiality agreement. We only use information related to the game state in our research, without any identity information. In addition, special equipment and accounts are provided to the participants to prevent leakage of equipment and account information during the test. The identity information of all participants is not disclosed to the public.
>     - Finally, the purpose of all tests is only for academic research. Our research will make contributions to the research community, the game industry, and the esports community.
>
> Q2: The potential societal consequences.
>
> - To the research community. MOBA poses a grand challenge to the AI community. We would believe that the study of the MOBA games will become the next AI milestone like AlphaGo or AlphaStar. However, the environment between MOBA games and the real world is very different. Different environments represent different tasks, and our method is only suitable for MOBA game-playing, just like study on other games such as Atari and Go. Therefore, our method cannot be applied to the military. We herewith expect this work to provide inspiration to other game-playing problems.
>
> - To the game industry. Our AI has found several real-world applications in the game, and is changing the way that MOBA game designers work, elaborated as follows: 1) Game balance testing. In MOBA and many other game types, balancing the ability of each strategy is essential. Using similar techniques presented in this paper is an easy way to construct a strategy balance testing tool for MOBA games. 2) PVE (player vs environment) game mode. Introducing AI with diverse strategies into the game's PVE mode is a low-cost method to increase the interest of players.
>
> - To the esports community. Like AlphaGo, our method can provide a high-quality low-cost training environment with diverse strategies for e-sports players.

---

### Review · Ethics_Reviewer_xVjd · 2021-08-17

**Recommendation:**

I recommend that the authors explicitly address the lack of consent, the (ostensible) lack of an IRB, and the de-identified nature of the detail in as much detail as possible in the next version of the paper. I also recommend they bring the answers on the Author Checklist into alignment with the current version of the paper by changing/adding to either the draft of the paper or the details reported on the checklist.

**Ethical Issues:**

Yes

**Ethics Review:**

The paper reports on a learning paradigm to train diverse policies in a specific video game by using human data to extract future states as goals. The reviewers note ethical concerns around human data subject consent, the possible need for an IRB, and the gaps between the way the authors respond to the Author Checklist and study details as reported in the current version of the paper.

As Reviewer S81z points out, there may be larger gaps between the way that the authors responded to the Author Checklist and the details in this version of the paper: "Lack of error bars: Figures 6, 9 and Tables 1,2,3 lack error bars, despite the authors having answered “Yes” to the checklist question about whether they have included error bars."

To be accepted, I recommend the authors bring the author checklist into alignment with the current draft by adding additional details to the draft paper.

---

> ### Author Response · Authors · 2021-08-30
> **Response to Ethics Reviewer xVjd**
>
> Thank you for your suggestion. We will add relevant explanations and modifications in the updated version.
>
> Q1：About CONSENT.
>
> A1: As a game activity, we will clearly indicate that this is an activity to challenge AI at the “entry” of the activity. In addition, near the "Start Challenge" button, we will also describe the number of in-game currency rewards that can be obtained by participating in the activity. These in-game currencies are usually used to redeem items in the game. The activity has no effect on the normal game process. This means that players can voluntarily choose whether to participate in the activity. This activity is extra entertaining for human players. We only use the result information of the match in our research, without any identity information.
>
> Q2：About IRB and IDENTIFIABILITY.
>
> A2: We misunderstood the meaning of IRB in the Checklist. "Honor of Kings" is a released and recognized "testbed" for MOBA-game-playing problems. Ye et al.[2020b,c,a] and Wu [2019] both used "Honor of Kings" as the "testbed" for research, including tests with top players. We consulted them and implemented similar measures and agreements to protect the welfare, rights, and privacy of human subjects.
>
> We have performed a process similar to IRB before the test conducted in this paper. Our research has been approved by our institution and all participants.
>
> - First, we analyze the risks of these experiments to the participants. The risks mainly include the leakage of identity information and the time cost.
>
> - Then, a series of measures are implemented to prevent these risks. We make a risk statement for participants and sign an identity information confidentiality agreement. We only use information related to the game state in our research, without any identity information. In addition, special equipment and accounts are provided to the participants to prevent leakage of equipment and account information during the test. The identity information of all participants is not disclosed to the public.
>
> - Finally, the purpose of all tests is only for academic research. Our research will make contributions to the research community, the game industry, and the esports community. (See A2 of Reviewer GhhQ)

---

### Decision · Program_Chairs · 2021-09-27

**Decision:**

Accept (Poster)

**Comment:**

After reading the reviews, authors response and discussions, I suggest to accept the paper. The ethical concerns have been answered and the authors took action to conform to what was required by the ethical review. Questions and some concerns on the method have been answered by the authors during the rebuttal and one reviewer upgraded its score from weak reject to a weak accept. The authors try to tackle the problem of learning in a real-time multi-player game which is extremely difficult. Even if the method is not theoretically sound and heavily rely on human knowledge, it is still a technical prowess to achieve good performance on such games. I will also suggest the authors to change their claim concerning the analysis of Fig 6 (b) where the difference between RL and their method is extremely minor.